# Tryptophan C-mannosylation is critical for *Plasmodium falciparum* transmission

Sash Lopaticki[1,4], Robyn McConville[1,2,4], Alan John [1,2,4], Niall Geoghegan [1,2], Shihab Deen Mohamed [1,2], Lisa Verzier[1,2], Ryan W. J. Steel [1,2], Cindy Evelyn [1], Matthew T. O'Neill [1], Niccolay Madiedo Soler [1,2], Nichollas E. Scott [3], Kelly L. Rogers [1,2], Ethan D. Goddard-Borger [1,2,5] ✉ & Justin A. Boddey [1,2,5] ✉

Tryptophan C-mannosylation stabilizes proteins bearing a thrombospondin repeat (TSR) domain in metazoans. Here we show that *Plasmodium falciparum* expresses a DPY19 tryptophan C-mannosyltransferase in the endoplasmic reticulum and that *DPY19*-deficiency abolishes C-glycosylation, destabilizes members of the TRAP adhesin family and inhibits transmission to mosquitoes. Imaging *P. falciparum* gametogenesis in its entirety in four dimensions using lattice light-sheet microscopy reveals defects in Δ*DPY19* gametocyte egress and exflagellation. While egress is diminished, Δ*DPY19* microgametes still fertilize macrogametes, forming ookinetes, but these are abrogated for mosquito infection. The gametogenesis defects correspond with destabilization of MTRAP, which we show is C-mannosylated in *P. falciparum*, and the ookinete defect is concordant with defective CTRP secretion on the Δ*DPY19* background. Genetic complementation of *DPY19* restores ookinete infectivity, sporozoite production and C-mannosylation activity. Therefore, tryptophan C-mannosylation by DPY19 ensures TSR protein quality control at two lifecycle stages for successful transmission of the human malaria parasite.

Tryptophan C-mannosylation is the only known form of protein C-linked glycosylation and is arguably the least well understood form of protein glycosylation[1]. This modification is installed in the endoplasmic reticulum (ER) by the DPY19 enzymes in metazoans: integral membrane proteins that use dolichol-mannose-phosphate as a glycosyl donor to form a C-glycosidic bond between D-mannose and tryptophan residues within the WxxW/C consensus motif[2–5]. Tryptophan C-mannosylation stabilizes a structural motif called the tryptophan ladder, which is most commonly found in proteins bearing a thrombospondin repeat (TSR) domain[6–8]. This domain almost universally contains a WxxWxxC motif, which is recognized and modified by DPY19 homologs[5,9]. Several studies have revealed that tryptophan C-mannosylation enhances the expression of proteins with a TSR domain: it stabilises the fold of these proteins to enable their secretion

to the cell surface in metazoans[7–10] and the Apicomplexan parasite *Toxoplasma*[11]. C-mannosylation can also play a role in TSR domain function[4,8,12,13].

TSR domains and tryptophan C-mannosylation by the DPY19 enzymes appear to have co-evolved during the emergence of metazoans. Apicomplexan protists are the only single-celled organisms known to produce proteins with TSR domains, including 'thrombospondin related anonymous protein' (TRAP) in *Plasmodium* spp.[14] and 'micronemal protein 2' (MIC2) in *Toxoplasma*[15]. These proteins may have been acquired early in Apicomplexan evolution through horizontal gene transfer with their metazoan hosts. This phylum includes parasites that cause significant human diseases. *Plasmodium* spp. are the etiological agents of malaria, and in 2020, they caused 241 million cases of this disease and 627,000 deaths[16].

[1]The Walter and Eliza Hall Institute of Medical Research, 1 G Royal Parade, Parkville, VIC 3052, Australia. [2]Department of Medical Biology, University of Melbourne, Parkville, VIC 3010, Australia. [3]Department of Microbiology and Immunology, University of Melbourne at the Peter Doherty Institute for Infection and Immunity, Parkville, VIC 3010, Australia. [4]These authors contributed equally: Sash Lopaticki, Robyn McConville, Alan John. [5]These authors jointly supervised this work: Ethan D. Goddard-Borger, Justin A. Boddey. ✉e-mail: goddard-borger.e@wehi.edu.au; boddey@wehi.edu.au

*P. falciparum* is responsible for the greatest mortality, predominantly in children under 5 years of age in sub-Saharan Africa.

The ten *Plasmodium* proteins of the TRAP adhesin family possess TSR domains and play essential roles in host cell infection during asexual[17], sexual[18,19], ookinete[20–22] and sporozoite[23–29] stages of the lifecycle. These proteins are expressed on the cell surface and, except for 'circumsporozoite protein' (CSP), are believed to provide a proteinaceous connection between the intracellular actomyosin motor and the extracellular environment to generate forces for parasite gliding motility, invasion and egress from host cells. Some TSR proteins are antimalarial vaccine candidates and CSP, including the TSR domain, is the target of the most advanced malaria vaccine, RTS,S/AS01 (Mosquirix™)[30].

C-glycosylation has been detected in sporozoites for TRAP in *P. falciparum*[31] and TRAP and CSP in *P. yoelii*[32,33]. Sporozoites are the infectious form of the parasite introduced by mosquitoes during blood feeding responsible for initiating mammalian liver infections. The significance of these modifications has not been explored, nor is it clear what other *Plasmodium* TSR proteins are similarly modified. Heterologous expression of *P. falciparum* DPY19 in mammalian cells deficient for C-mannosylation demonstrated it has C-mannosyltransferase activity for TRAP[34]. Genetic disruption of *DPY19* in *P. falciparum* caused no detectable phenotype in the asexual blood stage[35] therefore definitive proof of its function within the parasite is still lacking. We have previously shown that inhibition of *P. falciparum* TSR O-glycosylation by genetic disruption of *POFUT2* manifested in loss of protein stability and trafficking, as well as reduced parasite fitness during transmission to mosquitoes and infection of the liver in humanized mice[36]. Given the conservation and proximity of O- and C-glycosylation sites in TSR domains of *Plasmodium* proteins, we hypothesized that C-glycosylation may aslo be important in malaria parasites.

Here, we demonstrate that DPY19 is an ER-resident C-mannosyltransferase in malaria parasites. We provide evidence that 'secreted protein with altered thrombospondin repeat domain' (SPATR) and 'merozoite TRAP-like protein' (MTRAP) are tryptophan C-mannosylated like TRAP. We further show that genetic disruption of *DPY19* in *P. falciparum* is necessary and sufficient to prevent C-mannosylation in the parasite, impair egress following gametocyte activation and abrogate ookinete infection of the mosquito midgut. Complementation of *DPY19* restored the transmission defect and re-instated C-mannosylation activity, definitively validating tryptophan C-mannosylation as essential for *P. falciparum* infection of the mosquito, which is critical for onward transmission of the parasite.

## Results

### Identification of new domains in *Plasmodium* TSR proteins

The TRAP adhesin family comprises ten highly modular proteins with a TSR domain expressed throughout the *P. falciparum* lifecycle (Fig. 1a, b). Although many of these are well-studied proteins, domain assignments have largely been made on the sole basis of multiple sequence alignments (MSAs) and some limited structural data. To reaffirm these assignments, and possibly make new predictions, we performed DALI searches on globular domains predicted in the AlphaFold 2 models of all ten proteins that were available in the UniProt database. This provided independent verification of the current assignments and revealed that there are possibly hitherto unknown globular domains in some of these proteins (Fig. 1a, Supplementary Data 1). All of the proteins with a TSR domain possess a potential C-mannosylation motif WxxW/C (Fig. 1b) and there is evidence for C-hexosylation of TRAP in sporozoites[31,32], implying that a *Plasmodium* C-glycosyltransferase is expressed throughout the parasite's lifecycle. The single enzyme likely to perform this function, *P. falciparum* DPY19 (PF3D7_0806200)[31], shares 36.8% similarity and 13.1% identity with the well-studied *Caenorhabditis elegans* DPY19

(CCD62139.1)[5] (Supplementary Fig. 1). *P. falciparum* DPY19 contains 13 predicted transmembrane domains and is conserved across *Plasmodium* spp. (Fig. 1c). The DPY19 enzymes of *Plasmodium* spp. also retain a conserved glutamic acid residue that is important for enzymatic activity: E647 in *P. falciparum* DPY19, which corresponds with E579 in *C. elegans* DPY19 (Fig. 1d; Supplementary Fig. 1)[8].

### DPY19 expression in the *Plasmodium* ER

C-mannosylation of metazoan proteins is confined in the ER[4]. To determine the subcellular localization of DPY19 in *P. falciparum*, transgenic NF54 parasites were produced, whereby the *DPY19* gene was tagged at the C-terminus with a triple hemagglutinin (HA) epitope. Integration of the HA epitope was demonstrated by Southern blot (Supplementary Fig. 2) and successful tagging confirmed by immunoblot (Fig. 1e). Immunofluorescence microscopy revealed a perinuclear localization for DPY19-HA that co-localized with the ER-resident protease plasmepsin V[37] in asexual parasites (Fig. 1f). The distribution pattern for DPY19-HA indicates that this enzyme is most likely localized within the ER, similar to metazoans.

### Generation of *DPY19*-deficient *Plasmodium*

To study the role of DPY19, NF54 parasites were generated in which the *DPY19* gene was excised by double cross-over homologous recombination using 'clustered regularly interspaced short palindromic repeat' (CRISPR) / 'CRISPR-associated protein 9' (Cas9) gene editing (Fig. 2a). Two independent clones of Δ*DPY19* parasites (c1 and c2) were generated by independent transfection and validated by Southern blot analysis (Fig. 2b). Both mutant clones grew within human erythrocytes at the same rate as NF54 parasites, in agreement with previous work that observed no blood-stage defects upon disruption of *DPY19*[35] (Fig. 2c). POFUT2, which also glycosylates TSR domains, is expressed in *P. falciparum* blood stages and also exhibits no blood stage phenotype upon genetic disruption[36]. To exclude the possibility that POFUT2-mediated O-glycosylation and DPY19-mediated C-glycosylation compensate for the loss of each other in blood stages, we produced double mutant parasites. Two previously generated Δ*POFUT2* clones (D3 and G8)[36] were individually transfected with the *DPY19* knockout construct to excise the gene as described above. Δ*POFUT2*,Δ*DPY19* double mutant clones validated by Southern blot (Supplementary Fig. 3) were found, once again, to exhibit no defect in blood stage growth (Fig. 2c). Therefore, despite the co-expression and ER localization of these TSR-modifying glycosyltransferases in asexual blood stages, both the DPY19 and POFUT2 enzymes are dispensable for in vitro growth of *P. falciparum* in human erythrocytes under the conditions tested.

### DPY19 stabilizes *Plasmodium* TSR proteins

To probe the role of DPY19 in *Plasmodium* protein quality control, we investigated its impact on expression levels of the three TSR proteins expressed in blood stages (Fig. 1b). This included SPATR, '*Plasmodium* thrombospondin-related apical membrane protein' (PTRAMP) and MTRAP. PTRAMP and SPATR are involved in infection of erythrocytes[17,38]. MTRAP is expressed in asexual and gametocyte stages, is dispensable for asexual growth[18], and is necessary for lysis of the gamete-containing parasitophorous vacuole membrane (PVM) during egress within the mosquito midgut[18,19]. Immunoblotting and densitometric analysis indicated that levels of SPATR and MTRAP but not PTRAMP were decreased in Δ*DPY19* parasites (Fig. 2d). Therefore, DPY19 activity is necessary to maintain normal levels of SPATR and MTRAP but not all TSR proteins in *P. falciparum*. This is potentially because enhanced folding of the C-mannosylated TSR domain stabilizes the protein in parasites. The fold of PTRAMP may be inherently more stable than SPTAR and MTRAP, or it could be 'chaperoned' through associations with other proteins. The absence of a growth defect for Δ*DPY19* parasites despite a reduction in levels of the

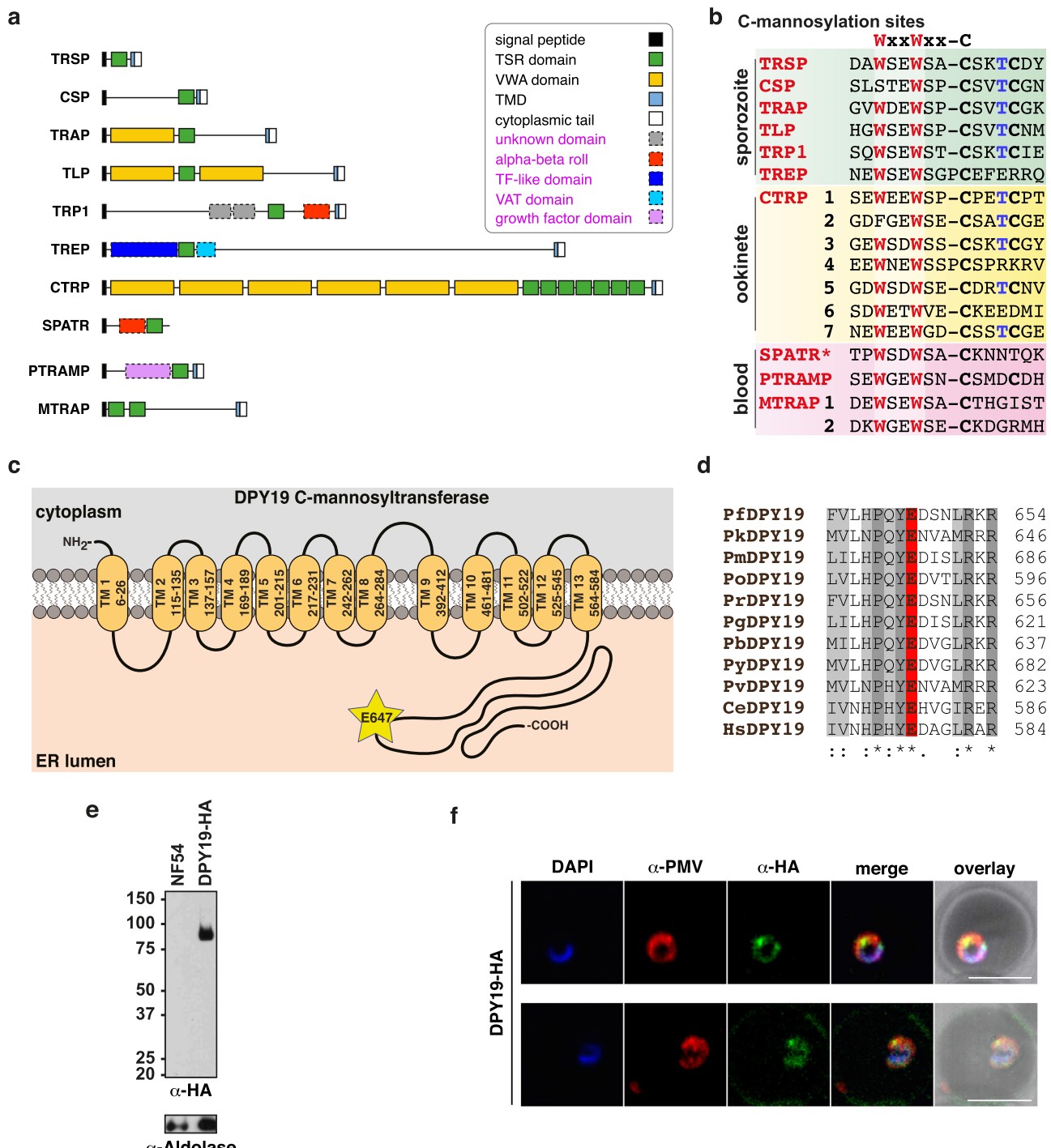

**Fig. 1 | TRAP adhesin family, putative C-mannosylation motifs and DPY19 in _Plasmodium falciparum_. a** A schematic illustrating the modular architecture of _P. falciparum_ proteins with a TSR domain. The five domain classes highlighted in purple text are new assignments made by performing DALI searches (Z > 2) on novel domains found within the Alphafold 2 models (UniProt) available for these proteins. **b** Multiple sequence alignment of TSR domain sequences from _P. falciparum_ showing the C-mannosylation consensus sequence WxxWxxC (red) and parasite lifecycle stage of expression. The O-glycosylated reside within the CxxS/TC motif (blue) is shown for reference. *SPATR is also expressed in _P. falciparum_ sporozoites. **c** Predicted topology of _P. falciparum_ DPY19 (PF3D7_0806200) including trans-membrane domains (TM) in the ER, predicted with TOPCONS. The conserved

glutamate required for catalytic activity in _C. elegans_ DPY19 is shown (E647). **d** Multiple sequence alignment of DPY19 in _Plasmodium_ spp., _C. elegans_ and _H. sapiens_, including the conserved glutamate required for catalysis in _C. elegans_ DPY19. **e** Western blot analysis of _P. falciparum_ NF54 and DPY19-HA schizont membrane extracts using antibodies to HA. Aldolase was included as a loading control. The same blot was probed consecutively with each antibody. Aldolase loading control is ~40 kDa. Shown is a representative of two blots. **f** Immunofluorescence microscopy of DPY19-HA ring stage parasites stained for the ER-resident marker protein plasmepsin V (PMV, red) and DPY19-HA (green) with nuclei visualized using DAPI (blue). Scale, 5 μm. Data is representative of _n_ = 2 independent experiments.

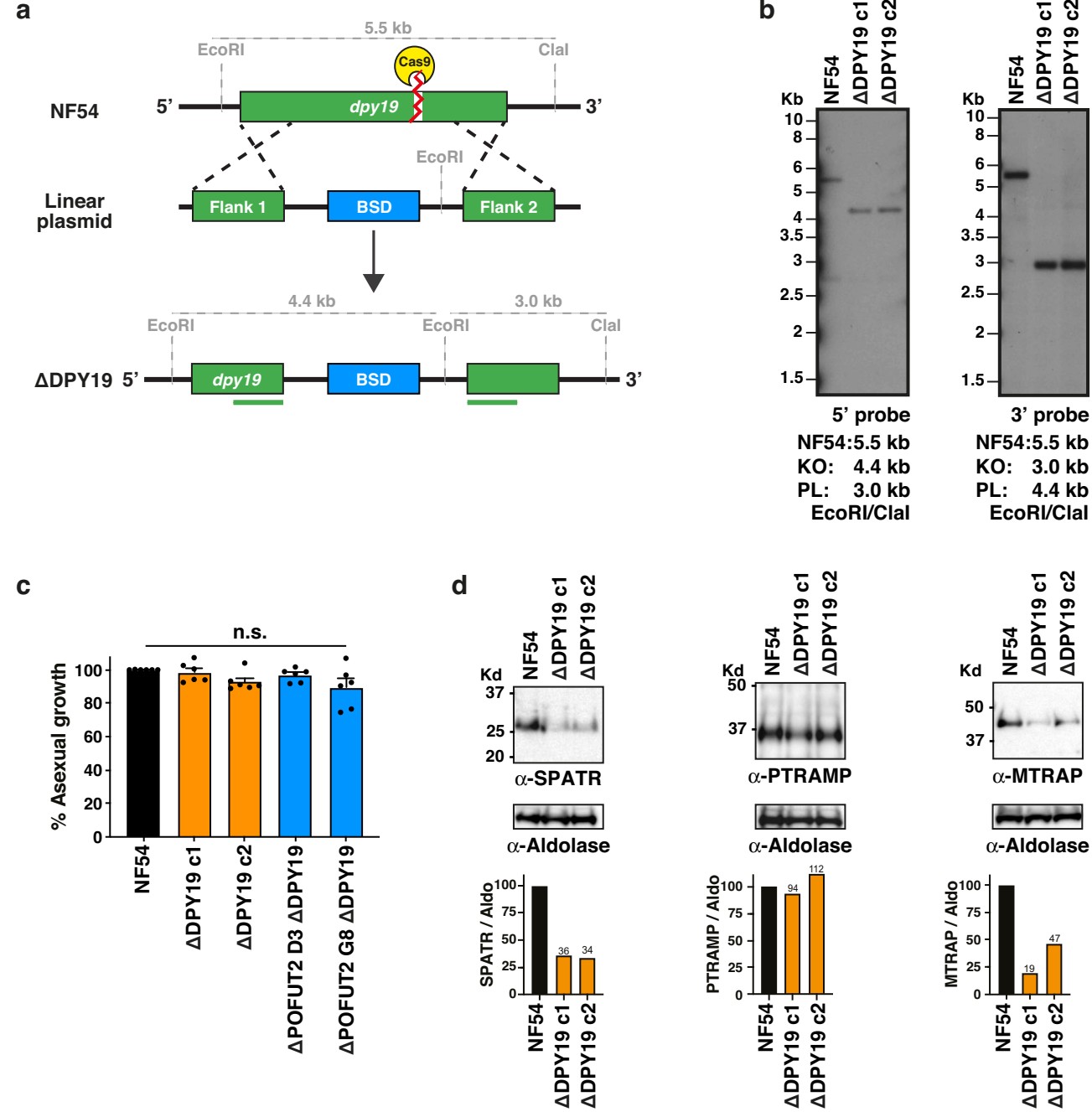

**Fig. 2 | *DPY19*-deficient *P. falciparum* have decreased levels of TSR proteins.**
**a** Strategy for disrupting the *DPY19* locus in *P. falciparum* NF54. The true 10 exon gene model is not depicted, for simplicity. **b** Southern blot analysis of parental NF54 and two clonal Δ*DPY19* clones c1 and c2, generated by independent transfections. Data is representative from *n* = 2 independent experiments. **c** Quantitation of the asexual growth rate of Δ*DPY19* (clones c1 and c2) cultures in human erythrocytes relative to NF54. Shown is pooled data from *n* = 6 independent experiments. Data are mean ± s.e.m. and were compared by one-way ANOVA (Kruskal-

Wallis test). **d** Western blot analysis of NF54 and Δ*DPY19* asexual schizonts using antibodies to SPATR, PTRAMP and MTRAP. Aldolase was used as a loading control. The same blot was probed consecutively with each antibody. Densitometry of each TSR protein band relative to aldolase loading control, and remaining protein in the C-glycosylation mutants, is indicated. Shown is a representative blot from two independent experiments showing the same result. Aldolase loading control is ~40 kDa.

essential protein SPATR[17] indicates that a decrease of this protein is tolerated by *P. falciparum* in vitro.

## SPATR and MTRAP are C-mannosylated

The dependence of SPATR and MTRAP expression levels on DPY19 in *P. falciparum* suggested that these proteins are C-glycosylated and that their folding and/or stability are enhanced by this modification. To examine this further, we produced C-glycosylated recombinant

SPATR and MTRAP (rSPATR and rMTRAP, respectively) for in vitro analyses of protein stability. *P. falciparum* SPATR and MTRAP were secreted as a fusion protein with an N-terminal FLAG-SUMO* tag by *Pichia pastoris* GS115 (a yeast with no C-mannosyltransferase) and a strain engineered to constitutively co-express the *C. elegans* C-mannosyltransferase (*Ce*DPY19)[8]. While the rMTRAP construct expressed well either with or without the co-expression of a C-mannosyltransferase, the rSPATR construct only expressed well in the

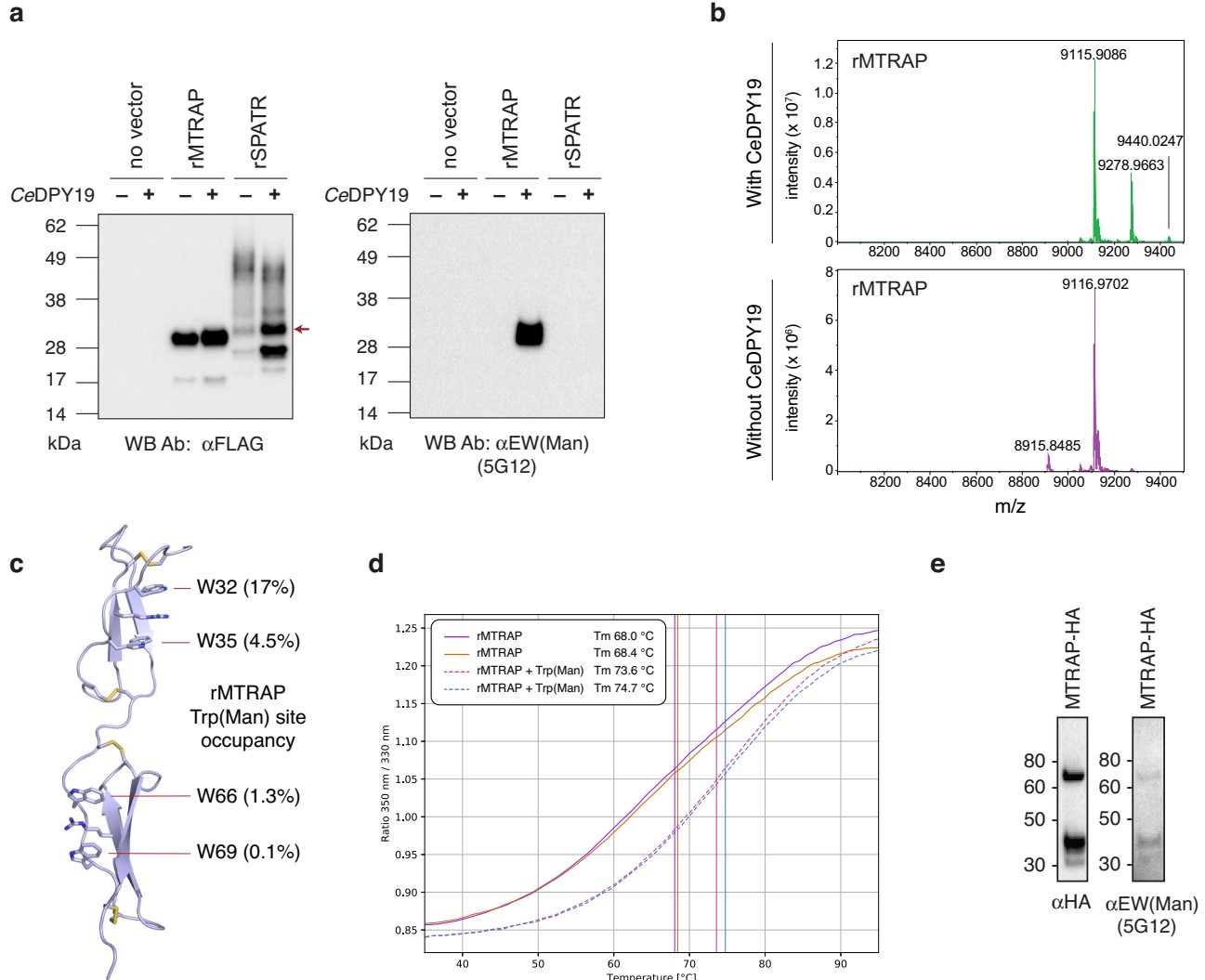

**Fig. 3 | Tryptophan C-mannosylation stabilizes MTRAP. a** Western blot of supernatant recombinant MTRAP (rMTRAP) and SPATR (rSPATR) expressed in yeast cells with and without co-expression of a C-mannosyltransferase (CeDPY19). The 5G12 primary antibody is specific for the EW(Man) motif found in MTRAP but not SPATR. Data is representative of $n = 2$ independent experiments. **b** Intact ESI-MS analysis of these rMTRAP samples following reduction and alkylation with iodoacetamide. **c** rMTRAP C-mannosylation site occupancy data mapped onto the TSR domains of the *P. falciparum* MTRAP AlphaFold 2 model (Uniprot: Q8IJB7). **d** Thermal denaturation profiles and Tm values for rMTRAP with (dashed lines) and without (hard lines) C-mannosylation, as determined by nanoDSF. Duplicate curves for each condition are shown. **e** Western blots for HA-tagged MTRAP affinity-purified from *P. falciparum* blood stage schizonts. The 5G12 primary antibody is specific for the EW(Man) motif found in the two TSR domains of MTRAP. Data is representative from $n = 1$ experiment.

presence of a C-mannosyltransferase (Fig. 3a). Immunoblotting these proteins with the monoclonal antibody 5G12, which is specific for tryptophan C-mannosylation and recognises the H/F/F/L/Q/EW(Man) epitopes[8,39], confirmed the C-mannosylation of rMTRAP when co-expressed with CeDPY19 (Fig. 3a). The rSPATR protein possesses a DW(Man) epitope and was therefore not recognized by the 5G12 mAb (Fig. 3a). Thus, to confirm that the rSPATR protein was modified by the C-mannosyltransferase, it was subjected to glycoproteomic analysis: the major species was overwhelmingly the mono-C-glycosylated form, with C-hexosylation occurring on W175 (Supplementary Fig. 4). This confirmed that C-glycosylation plays an important role in the folding and/or stability of SPATR, just as we had observed in *P. falciparum* (Fig. 2d). The rMTRAP protein expressed in the presence and absence of the C-mannosyltransferase was purified from the culture supernatant and the FLAG-SUMO* tag removed. The purified TSR domains were analysed by intact ESI-MS[8,39] comparisons of deconvoluted mass spectra revealed that the C-glycosylated sample possessed +162 Da and +324 Da proteoforms, commensurate with the addition of 1-2 hexoses (Fig. 3b). LC-MS/MS analysis of trypsin-digested samples of these proteins confirmed that this mass shift was due to C-mannosylation on the two WxxW motifs within MTRAP (Supplementary Fig. 5), and quantitation by LC/MS enabled an estimation of the occupancy at each site (Fig. 3c, Supplementary Fig. 6). Differential scanning fluorimetry (nanoDSF) was used to assess the stability of rMTRAP with and without C-glycosylation. The C-glycosylated samples exhibited a bulk Tm ≈6 °C higher than the unglycosylated samples (Fig. 3d), even though glycosylation occupancy in this sample was modest (≈20%) (Fig. 3c, SI Fig. 6). This indicates that C-mannosylation confers considerable additional stability to the MTRAP protein fold.

The 5G12 mAb was then used to confirm that C-mannosylation was occurring on MTRAP in *P. falciparum* blood stages. MTRAP-HA was affinity purified from transgenic *P. falciparum* using anti-HA agarose and the eluate analysed by immunoblot. HA antibodies specifically identified MTRAP-HA as two protein species, p58 and p35, as previously described[40]. Probing the same membrane with 5G12 antibodies identified that the same bands contained the modified EW(Man)

epitope(s), suggesting that tryptophan C-mannosylation occurs on MTRAP in *P. falciparum* blood stages (Fig. 3e). Altogether, these results show that C-mannosylation of *P. falciparum* SPATR and MTRAP stabilises their folding.

## DPY19 facilitates egress from erythrocytes following gametocyte activation

Gametocytes ingested by mosquitoes during a bloodmeal use MTRAP to lyse the PVM, allowing gametes to egress from the erythrocyte for sexual fertilization and onward transmission[18]. As we had established a direct link between DPY19 and MTRAP protein levels and stability, we investigated the function of C-mannosylation in sexual stages. ΔDPY19 parasites were differentiated into gametocytes and no difference in stage V gametocytemia was observed between parasite lines, indicating that DPY19 is not required for gametocytogenesis (Supplementary Fig. 7a). We did not detect expression of SPATR or PTRAMP in stage V gametocytes, therefore these TSR proteins are unlikely to have a function in mature gametocytes (Supplementary Fig. 7b), leaving MTRAP as the only known TSR protein expressed at this lifecycle stage.

Activation of microgametocytes and macrogametocytes in the mosquito midgut initiates the development of gametes that egress by an inside-out mechanism of PVM lysis followed by erythrocyte membrane rupture[41] for infection of the mosquito. Until very recently[42], only limited live imaging of *P. falciparum* gametogenesis has been reported[41,43] and capturing the whole process from activation through to complete exflagellation has remained challenging. To understand the function of C-mannosylation, we used lattice light-sheet microscopy to visualize the entirety of gametogenesis by both male and female NF54 and ΔDPY19 parasites in four dimensions. Gametocytes were labelled with SPY-tubulin to visualize microtubule dynamics and cell membranes were labelled with the dye, Di-4-ANEPPDHQ. Upon addition of xanthuric acid, NF54 microgametocytes shifted from falciform to a round shape indicating activation had occurred, and evidence of mitotic spindle formation was visible by a single punctate region of SPY-tubulin labelling (Fig. 4a). With NF54, we witnessed microgametogenesis proceed to exflagellation via two pathways: (i) full gametocyte egress from the erythrocyte following activation, with axoneme assembly and microgamete exflagellation originating from the extracellular parasite (44% frequency; Fig. 4a, h, Supplementary Mov. 1), and (ii) partial gametocyte egress involving axoneme assembly within an intracellular parasite, subsequent erythrocyte membrane unwrapping with partial gametocyte egress, and exflagellation occurring from the partly intracellular residual body (44% frequency; Fig. 4b, h, Supplementary Mov. 2). A small number of NF54 microgametocytes assembled axonemes but exflagellation failed (12% frequency; Fig. 4d, h, Supplementary Mov. 3). In contrast, while all ΔDPY19 microgametocytes became rounded after activation, 79% of gametocytes did not egress from the erythrocyte during gametogenesis. The composition of these events included: 29% of gametocytes for which axoneme assembly within the intracellular gametocyte caused visible protrusions of the enclosed red blood cell membrane followed eventually by explosive exflagellation with the gametocyte residual body remaining inside the erythrocyte (Fig. 4c, h, Supplementary Mov. 4). The other 50% of gametocytes that did not egress the erythrocyte involved spindle formation and axoneme assembly but failure of exflagellation (Fig. 4e, Supplementary Mov. 5). The remaining ΔDPY19 events were similar to NF54, resulting in exflagellation (14% full gametocyte egress, 7% partial gametocyte egress; Fig. 4h). Inspection and quantification by immunofluorescence and light microscopy, respectively, confirmed the presence of exflagellation, albeit at a reduced rate for ΔDPY19 parasites compared to NF54 consistent with the lattice light-sheet results (Supplementary Fig. 7c, d).

Female gametocytes undertake gametogenesis differently to males[41,43]. Imaging with lattice light sheet microscopy revealed that all NF54 and ΔDPY19 macrogametocytes switched from falciform to

round after activation however fewer ΔDPY19 macrogametocytes egressed the erythrocyte compared to NF54 (Fig. 4f, g, i, Supplementary Mov. 6, 7) demonstrating that DPY19 is important for egress by females, as well as males. Analysis of successful gametogenesis events by NF54 and ΔDPY19 revealed no significant differences in timing of key developmental events between parasite lines (Fig. 4j). Collectively, lattice light-sheet imaging revealed that both males and females use DPY19 to egress the erythrocyte during gametogenesis and that exflagellation can originate from a residual body located outside or sometimes inside the host erythrocyte.

## DPY19 is essential for transmission to mosquitoes

Successful gamete egress from erythrocytes permits sexual fertilization, producing ookinetes over the subsequent 20-36 hours[44] that invade the mosquito midgut. Ookinetes employ gliding motility to invade and traverse the midgut epithelium, which requires 'circumsporozoite- and TRAP-related protein' CTRP[20,22]. Once at the basal lamina, ookinetes differentiate into oocysts, within which sporozoites develop. To understand the function of C-mannosylation in ookinetes, mature gametocytes were fed to female *Anopheles stephensi* mosquitoes and ookinetes were isolated by dissecting midguts 23-32 hours post-bloodmeal. Qualitative inspection by immunofluorescence microscopy revealed the presence of mature ookinetes with generally similar morphology and quantification of mosquito extracts by light microscopy revealed no difference in the number of NF54 and ΔDPY19 ookinetes at multiple time points (Fig. 5a–c). This indicates that DPY19 is not essential for ookinete development. However, oocyst development at the basal lamina of mosquito midguts was abrogated for ΔDPY19 and ΔPOFUT2,ΔDPY19 double mutants relative to NF54, as was the prevalence of mosquito infection (Fig. 5d), indicating that DPY19 is essential for transmission. Genetic complementation of the *DPY19* gene into ΔDPY19 parasites using CRISPR/Cas9 (Supplementary Fig. 8) restored oocyst production and mosquito infection prevalence at two different transmission densities (Fig. 5e). Complementation also reinstated salivary gland sporozoite production as a result of normal oocyst intensities (Fig. 6a). These results definitively show that DPY19 is critical for ookinete infection of the mosquito.

To better understand the dynamics of ookinete infection, mature gametocytes were fed to mosquitoes and quantitative PCR was performed to measure expression of parasite and mosquito genes the day after blood feeding. Expression of *P. falciparum* genes *Pfs25* and *CTRP* was similar across conditions, indicating approximately equal numbers of gametocytes, zygotes and ookinetes were present in the midguts after feeding (Supplementary. Fig. 7e). LL3 is a mosquito transcription factor involved in regulating the immune response to infection with pathogens including *Plasmodium* ookinetes[45] and SRPN6 is an immune-responsive protease inhibitor activated in response to infection including with ookinetes[46]. Expression of LL3 and SRPN6 is upregulated following ookinete invasion of the epithelium in situ, providing molecular markers of ookinete invasion[45,46]. Relative to uninfected sugar-fed mosquitoes, An. stephensi midguts containing NF54 parasites had significantly increased *LL3* and *SRPN6* expression, confirming ookinete invasion of the midgut as expected. However, both ΔDPY19 clones failed to induce the same level of *LL3* or *SRPN6* expression as NF54 parasites (Fig. 5f). This suggests that DPY19-deficient ookinetes are not able to invade the midgut epithelium as efficiently as NF54 or that mosquito immune responses cannot recognize these mutant ookinetes effectively.

CTRP is the only known TSR protein expressed in ookinetes and is likely linked to the ΔDPY19 transmission phenotype because it contains seven tandem TSR domains (Fig. 1a, b). To further investigate this hypothesis, we expressed a chimeric protein comprised of a signal peptide and the seven tandem TSR domains of CTRP fused to green fluorescent protein in *P. falciparum* (CTRP-GFP). CTRP-GFP expression was low by microscopy, however, as DPY19 is expressed in the ER of

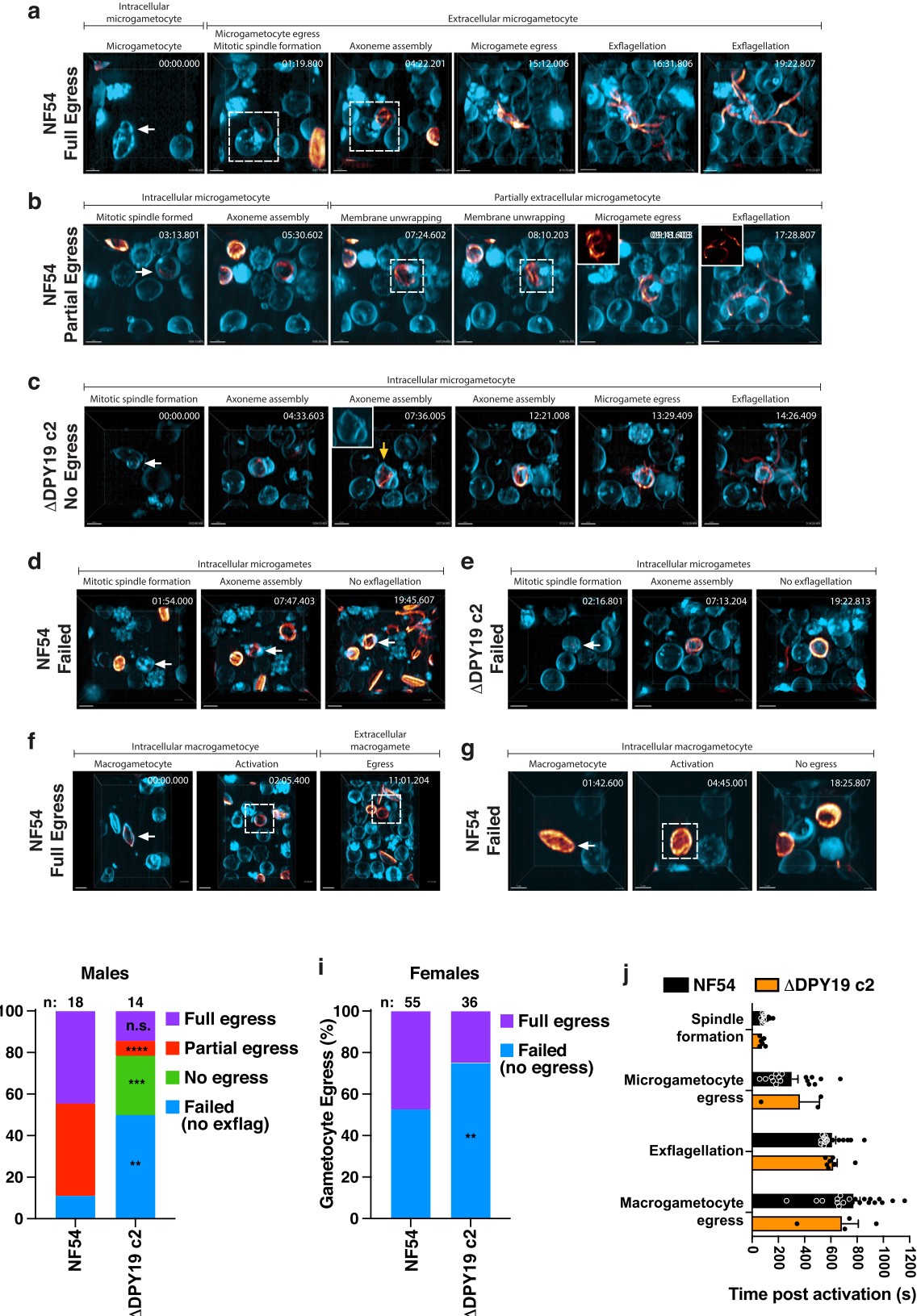

asexual stages, we used the asexual parasite cell as a surrogate system to monitor post-translational modifications in the ER and subsequent trafficking using immunoblotting with anti-GFP antibodies. Immunoblots detect key steps in *P. falciparum* protein secretion, including (i) processing of the N-terminal signal peptide following ER entry[47,48] and (ii) subsequent secretion of GFP chimeras through the endomembrane

system including to cytostomal vacuoles[49], resulting in degradation to a protease-resistant 'GFP core' in the food vacuole[47,48,50–52]. In NF54, CTRP-GFP migrated consistent with proteolytic cleavage of the ER-type signal peptide (~86 kDa) and secretion of the chimera was evident as a 'GFP core' band (~26 kDa) as expected (Fig. 5g). In contrast, CTRP-GFP expression in ΔDPY19 parasites resulted in accumulation of a

**Fig. 4 | Lattice light-sheet microscopy of *P. falciparum* gametogenesis to completion in 4D. a** Representative of NF54 involving full gametocyte egress. This includes microgametocyte (arrow) activation and egress (hashed box), spindle formation, axoneme assembly and exflagellation. Shown is a single representative of *n* = 8 similar events. Scale, 5 µm. **b** Representative of NF54 involving partial gametocyte egress. This includes microgametocyte (arrow) activation, spindle formation, axoneme assembly, erythrocyte membrane unwrapping (hashed box) and exflagellation. Shown is a single representative of *n* = 8 similar events. Scale, 4 µm. **c** Representative of ΔDPY19 involving no gametocyte egress with exflagellation from the residual body inside the erythrocyte. This includes microgametocyte (arrow) activation, spindle formation, axoneme assembly causing erythrocyte membrane protrusions (yellow arrow, inset) and exflagellation. Shown is a single representative of *n* = 4 similar events. Scale, 4 µm. **d** Representative of NF54 involving failed exflagellation. This includes microgametocyte (arrow) activation, spindle formation, axoneme assembly, no exflagellation. Shown is a single representative of *n* = 2 similar events. Scale, 5 µm. **e** Representative of ΔDPY19 involving failed exflagellation. This includes microgametocyte (arrow) activation, spindle formation, axoneme assembly, no exflagellation. Shown is a single representative of

*n* = 7 similar events. Scale, 5 µm. **f** Representative of NF54 involving macro-gametocyte full egress. This includes macrogametocyte (arrow) rounding up after activation (hashed box) and egress of the from the erythrocyte (hashed box). Shown is a single representative of *n* = 20 similar events. Scale, 5 µm. **g** Representative of NF54 involving no macrogametocyte egress. This includes macrogametocyte (arrow) rounding up after activation (hashed box) and no egress. Shown is a single representative of *n* = 26 similar events. Scale, 3 µm. **h** Quantification of full egress (purple), partial egress (red), no egress (green) or failed (no exflagellation; blue) by microgametocytes from lattice light-sheet imaging. Data were compared using Qi-Square analysis (Fisher's exact test one-tailed). ****$P$ = 0.0043, ***$P$ = 0.0276, **$P$ = 0.0225, n.s., not significant. Number of events (*n*) is shown. **i** Quantification of full (purple) or failed (blue) egress of macro-gametocytes from lattice light-sheet imaging. Data were compared using Qi-Square analysis (Fisher's exact test one-tailed) **$P$ = 0.0470. Number of events (*n*) is shown above each condition. **j** Quantification of successful egress events (seconds, s) by NF54 and ΔDPY19. Data are mean ± s.e.m. from a total of 43 events (NF54: 16 males, 20 females; ΔDPY19: 3 males, 4 females) compared by one-way ANOVA (Kruskal-Wallis test). No significant timing differences occurred between NF54 and ΔDPY19.

---

larger, possibly full-length, chimera, suggesting retention of the signal peptide and 'GFP core' was barely detectable, altogether indicating that CTRP-GFP was not secreted from the ER normally in the absence of DPY19 (Fig. 5g). Collectively, our results demonstrate that ookinetes require DPY19 to infect the mosquito midgut and provide evidence that this is because DPY19 facilitates CTRP secretion from the ER. Therefore, C-mannosylation may facilitate invasion and possibly traversal of the midgut epithelium by *P. falciparum* ookinetes.

## C-mannosyltransferase activity in sporozoites is DPY19-dependent

To study the function of DPY19 in sporozoites, we isolated parasites from salivary glands of infected mosquitoes. The number of salivary gland sporozoites was severely reduced in mosquitoes fed ΔDPY19 parasites compared to NF54 (Fig. 6a), an expected phenotype based on the oocyst defect described above. To investigate infectivity, sporozoites were incubated with human HC-04 hepatocytes in the presence of FITC-dextran followed by labelling intracellular sporozoites with anti-CSP antibodies and quantifying hepatocyte traversal and invasion by flow cytometry. No major difference in the rates of cell traversal or invasion were observed for ΔDPY19 sporozoites compared to NF54 (Supplementary Fig. 9, 10). Visualization of sporozoites by immunofluorescence microscopy with 5G12 antibodies detected the EW(Man) C-mannosylated epitope in NF54 and ΔDPY19 complemented sporozoites (ΔDPY19 Comp) as punctate signals that were not detectable in ΔDPY19 sporozoites (Fig. 6b, Supplementary Fig. 11). Immunoblot of salivary gland sporozoites with 5G12 antibodies identified at least one sporozoite-specific band in NF54 at ~70 kDa; this migrated larger than CSP, which is not C-glycosyated[31], and the same size as TRAP (Fig. 6c–e, arrow) in agreement with this protein being C-mannosylated[31]. All sporozoite-expressed TSR proteins with the exception of SPATR[53] contain the EW motif, which is compatible with 5G12 recognition should C-mannosylation occur (Fig. 1b). CSP is O-glycosylated but not C-glycosylated[31] and we did not detect any major difference in surface expression of CSP by immunofluorescence microscopy (Fig. 6b, Supplementary Fig. 11) or in protein levels by immunoblot (Fig. 6e) in ΔDPY19 sporozoites. Other 5G12-reactive sporozoite proteins were evident by immunoblotting but their intensity was weaker than for TRAP, which is known to be an abundant sporozoite protein (Fig. 6c). Interestingly, at least one mosquito-specific protein was detected by 5G12 (Fig. 6c, asterisk), indicating *An. stephensi* also performs C-mannosylation. Anti-TRAP antibodies detected this protein in NF54 and ΔDPY19 clones at similar expression levels, however, the C-mannosylated NF54 TRAP band recognized by 5G12 was absent from ΔDPY19 clones (Fig. 6d, e). Collectively, these

results establish DPY19 as a C-mannosyltranferase in *P. falciparum* whose expression in sporozoites coincides with tryptophan C-mannosylation of sporozoite proteins including TRAP.

## Discussion

The previous identification of O- and C-glycosylation of TRAP in *P. falciparum*[31] and *P. vivax*[32] sporozoites addressed a longstanding question in the malaria field by validating that *Plasmodium* parasites do glycosylate their proteins. Since then, independent laboratories have contributed to understanding the significance of glycosylation in different *Plasmodium* spp. Inactivation of the O-glycosylation pathway in *P. berghei* caused no phenotype in any lifecycle stage[54] and it is possible that the remaining C-glycosylation pathway contributed to stabilizing TSR proteins needed for lifecycle progression. Conversely, disruption of O-glycosylation in *P. falciparum* exposed a transmission phenotype during ookinete infection of the mosquito, albeit mild, and more pronounced phenotypes in sporozoites, at least in part due to destabilization, reduced expression, and impaired trafficking of TRAP[36]. The presence of the C-glycosylation consensus motif WxxW/C in all *P. falciparum* TSR domains alluded to tryptophan C-mannosylation also being important during the parasite lifecycle. Identification and heterologous expression of the *P. falciparum* DPY19 homolog demonstrated that it has catalytic activity[34] yet its genetic disruption in *P. falciparum* revealed no phenotype in asexual stages[35]. Thus, the significance of tryptophan C-mannosylation to malaria biology and definitive proof of the enzyme responsible has remained unknown. In this study, we showed that DPY19 is necessary for the stability and protein expression levels of SPATR and MTRAP and for secretion of CTRP from the ER in *P. falciparum*, although PTRAMP, CSP and TRAP were not dependent on DPY19. SPATR and PTRAMP are essential for asexual blood stage growth[17,38] and the levels of these TSR proteins in ΔDPY19 parasites was sufficient to sustain asexual growth in the absence of C-mannosylation. However, two subsequent *P. falciparum* lifecycle stages were clearly impacted by loss of DPY19 function: gametogenesis and ookinete infection of the mosquito midgut, both of which are essential for transmission.

Lattice light-sheet microscopy provided exquisite detail of gametogenesis from activation through to successful exflagellation in four dimensions. Our analyses revealed that microgametogenesis can have two outcomes, such that the gametocyte either fully egresses from the erythrocyte soon after activation, or it remains partially inside the erythrocyte for the remainder of gametogenesis whilst unwrapping the host cell membrane. A recent detailed live-imaging study of microgametogenesis in 4D described a similar partial egress of the microgametocyte, though they did not observe the parasites

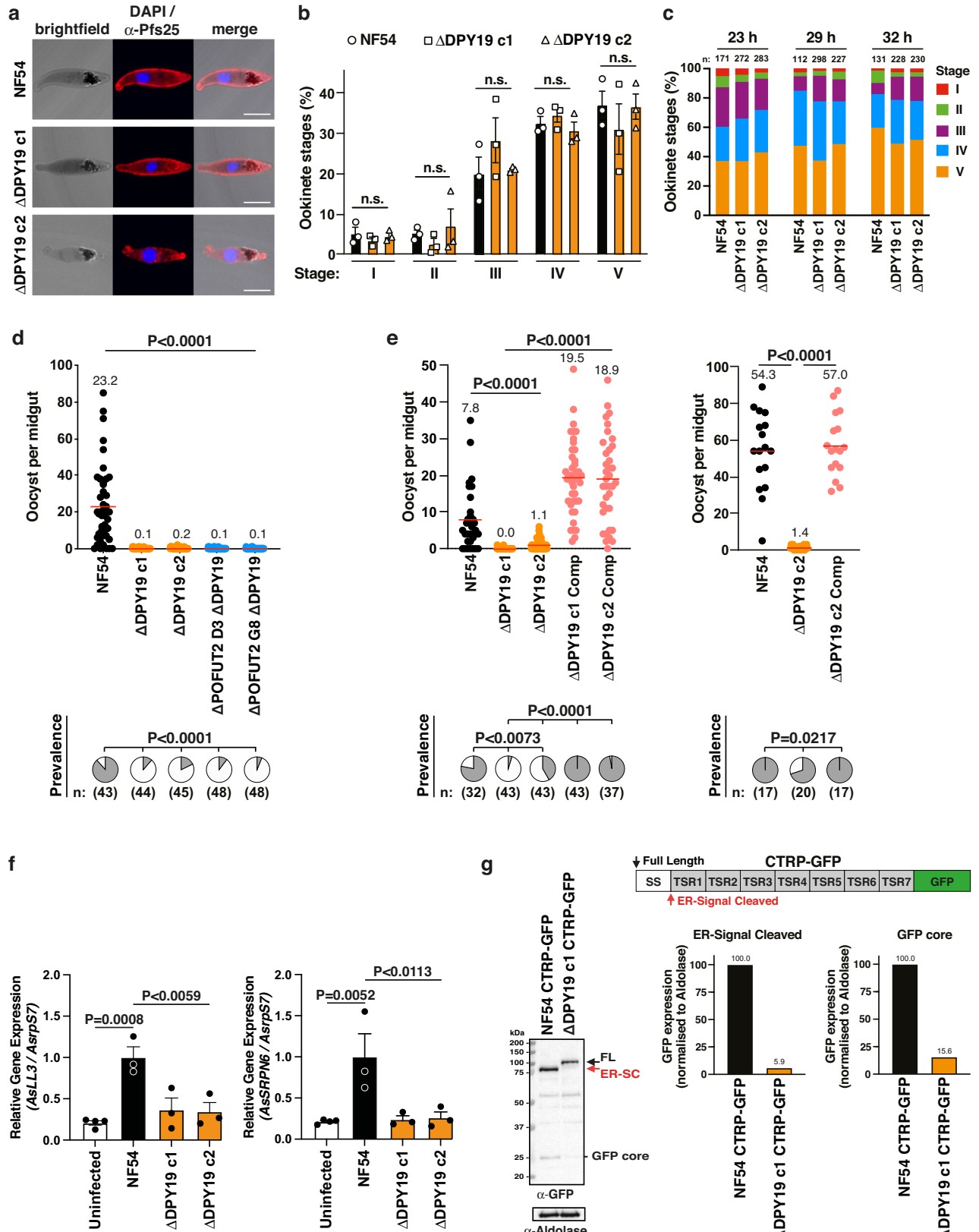

exflagellating, defining these events as incomplete[42]. In our study, all partially egressed gametocytes successfully completed exflagellation, suggesting that the beating forces of exflagellation may contribute to egress. The discrepancy in observing exflagellation may be due to the phototoxicity caused by the type of fluorescence microscopy method used in the recent study[42]. Lattice light-sheet microscopy is gentle and

allows biological processes to be observed, relatively unperturbed and at higher spatio-temporal resolution, for longer periods of time than traditional fluorescence microscopy[55,56]. This approach also allowed us to identify that *DPY19*-deficient gametes could not egress the erythrocyte normally, revealing that C-mannosylation functions to enable these processes. Gamete egress from the erythrocyte involves PVM

**Fig. 5 | DPY19 is essential for *P. falciparum* transmission to mosquitoes.**
**a** Immunofluorescence microscopy of *P. falciparum* ookinetes dissected 23 hours post bloodmeal with anti-Pfs25 antibody and DAPI. Scale, 5 μm. **b** Quantification of NF54 and Δ*DPY19* ookinetes in *An. stephensi* mosquitoes per parasite strain 23 hours post bloodmeal. Data are mean ± s.e.m. from *n* = 3 independent experiments analyzed by one-way ANOVA (Kruskal-Wallis test). n.s., not significant. **c** Quantification of ookinetes in *An. stephensi* guts per parasite strain over time post bloodmeal. Ookinetes counted (*n*:) are indicated. Shown is a single representative of *n* = 3 independent experiments showing similar results. **d, e** Mosquito oocyst intensity (top) and infection prevalence (bottom) following infection. Mean oocyst number is indicated and with a red bar within each condition. Mosquito sample size (*n*:) is shown. Oocysts were compared using Kruskal-Wallis one-tailed test with Dunn's correction and prevalence compared using the chi-square test (Fisher's exact test). Data are a single representative from 3 independent experiments. The two experiments in **e** show effect on transmission between parasite lines at lower (left) and higher (right) forces of infection (different oocyst burdens). *P* values are indicated. **f** RT-qPCR quantification of *An. stephensi LL3* and *SRPN6* mRNA expression relative to *An. stephensi rps7* in mosquitoes that were sugar fed (uninfected) or fed NF54- or Δ*DPY19*-infected blood. Data are mean ± s.e.m. from *n* = 25 midguts per condition per experiment from 3 independent experiments, pooled and analyzed by one-way ANOVA using Dunnet's test. *P* values are indicated. **g** Immunoblotting with anti-GFP antibodies shows DPY19-dependent inhibition of CTRP-GFP processing in schizonts and lack of the secreted GFP core. Schematic shows CTRP-GFP expressed from *PTRAMP* promoter to correspond with DPY19 expression. Full length protein (black arrow), ER-type signal sequence-cleaved protein (red arrow), secreted and degraded chimera in food vacuole (GFP core) are labeled. Aldolase loading control is ~40 kDa. Densitometry of the ER-type signal sequence-cleaved band and GFP core band relative to aldolase loading control is indicated for NF54 and Δ*DPY19* schizonts. Shown is a representative blot from two independent experiments showing the same result. SS, signal sequence; TSR, thrombospondin repeat domain, GFP, green fluorescent protein.

lysis soon after activation followed later by rupture of the red cell membrane from one location[41]. Our labelling approach did not allow us to follow lysis of the PVM specifically and so it was not evident when this occurred. Erythrocyte membrane protrusions were visible during axoneme assembly by Δ*DPY19* microgametocytes suggesting that the PVM may have lysed in those cells. Exflagellation occurred thereafter, from inside the erythrocyte at approximately the same time as NF54 exflagellation, supporting the idea that pulsating forces of exflagellation contribute to egress of microgametes. MTRAP contains two N-terminal TSR domains and is required for female and male gamete egress and exflagellation for transmission[18]. We have shown that tryptophan C-mannosylation of MTRAP by DPY19 stabilizes this protein and that activated males and females lacking this glycosyltransferase are defective for erythrocyte egress. It is likely that *P. falciparum* gametocytes lacking the C-mannosylation pathway possess less stably folded MTRAP, leading to defective egress after activation; however, remaining levels of MTRAP allowed gametogenesis to be completed in a subset of *DPY19*-deficient parasites. Interestingly, gamete egress by Δ*DPY19* did not completely phenocopy that reported for Δ*MTRAP*[18]. The different phenotypes are probably due to partial knockdown of MTRAP in Δ*DPY19*, compared to complete loss of this protein in Δ*MTRAP* parasites[18]. Whether MTRAP is solely responsible for the Δ*DPY19* egress phenotype or if other TSR proteins contribute is unclear, though we did not detect expression of SPATR or PTRAMP in stage V gametocytes, suggesting they are not participants in gamete egress. Loss of DPY19 expression did not apparently affect the formation or development of ookinetes in mosquitoes, suggesting that the egress defects observed did not contribute considerably to the transmission phenotype in *An. stephensi*, underscoring highlighting the significance of the Δ*DPY19* ookinete defect.

DPY19-deficient gametocytes fed to mosquitoes produced normal numbers of ookinetes however they had a severe defect in the production of oocysts. Ookinetes disrupted in C-mannosylation did not activate the mosquito midgut immune responses in the same way as NF54 ookinetes, consistent with an ookinete invasion or motility defect. In fulfilment of molecular Koch's postulates[57], genetic complementation of *DPY19* using CRISPR/Cas9 to target the Δ*DPY19* locus and stably restore gene expression re-established ookinete transmission and sporozoite production to wild-type levels whilst also reinstating tryptophan C-mannosylation activity in the parasite. This is to our knowledge the first use of CRISPR/Cas9 complementation to restore ookinete transmission in *P. falciparum*, highlighting a useful approach to mitigate inadvertent genetic mutations being misassigned to transmission defects in *P. falciparum*.

CTRP is an ookinete adhesive micronemal protein with seven tandem TSR domains, each possessing O- and/or C-glycosylation consensus motifs, strongly suggesting it is glycosylated. *P. falciparum* mutants lacking O-fucosylation[36] or C-mannosylation (this

study) are defective for oocyst production in mosquitoes. We propose that the tandem TSR domains of CTRP require glycosylation to fold correctly for secretion to the ookinete surface for midgut invasion, a hypothesis supported by defective CTRP-GFP secretion from the ER in *DPY19*-deficient *P. falciparum* in this study. The degree to which oocyst production was abrogated suggests that C-mannosylation is more important than O-fucosylation for the folding and trafficking of CTRP specifically. The *CTRP* gene has been genetically disrupted in *P. falciparum* and this prevented oocyst development[22], although whether ookinetes were specifically prevented from invading the epithelium was not determined. In *P. berghei*, *CTRP* disruption prevented ookinete invasion of the midgut epithelium[20,21] and our results suggest that CTRP is likely to be involved at this step in *P. falciparum*. Genetic removal of the TSR domains from *P. berghei* CTRP did not affect ookinete transmission whereas the von Willebrand factor A-like (vWA) domains were essential[58]. This could be interpreted as contradictory to our results. However, we propose that CTRP possessed improperly folded TSR domains when *DPY19* was disrupted, which interfered with secretion of the entire protein including the vWA domains, inhibiting infection. The presence of tandem TSR domains in CTRP of all *Plasmodium* species indicates there is likely a reason for their presence. A similar tandem TSR domain architecture is present in MIC2 of *T. gondii*, which act as a rigid stalk[59] and mediate the formation of a hexameric complex[60]; perhaps a similar phenomenon occurs in *Plasmodium* for CTRP.

CSP is not C-glycosylated in *P. falciparum* sporozoites[31], consistent with its TSR domain possessing a SxxWxxC motif rather than a WxxWxxC consensus sequence (Fig. 1b). We did not identify C-mannosylation of CSP by Western blot or any defect in CSP levels or localization by microscopy on the Δ*DPY19* background, consistent with this protein not being modified by DPY19. Earlier demonstration of C-glycosylation of TRAP[31,32] brought into question the functional significance of this modification in sporozoites, though this has previously remained unexplored. Loss of DPY19 activity abrogated the detection of tryptophan C-mannosylation in salivary gland sporozoites but did not affect TRAP protein levels, hepatocyte traversal or invasion in vitro. Conversely, loss of protein O-glycosylation destabilized TRAP protein levels significantly and impaired its trafficking in salivary gland sporozoites, reducing their infectivity[36]. This demonstrates that TRAP preferentially requires O-fucosylation over C-mannosylation for function during infection. These differences, together with SPATR and MTRAP requiring C-mannosylation but not O-fucosylation for stability in our study, highlight the protein-specific dependencies for O- and/or C-glycosylation in *P. falciparum*, which is commensurate with a similar conclusion in metazoan studies[61]. Therefore, C-glycosylation is dispensable in *P. falciparum* salivary gland sporozoites because O-fucosylation of essential TSR

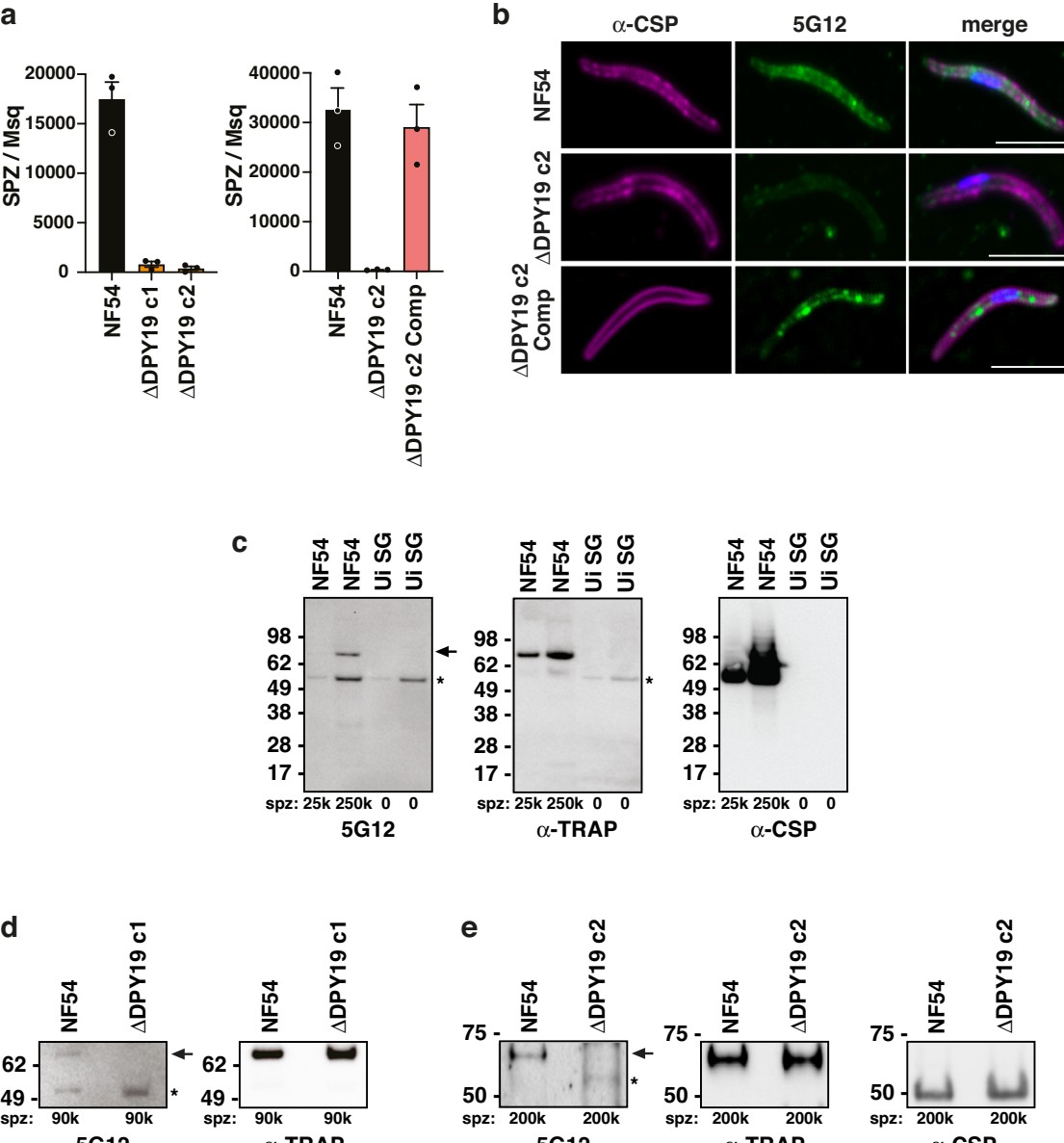

**Fig. 6 | DPY19-dependent C-mannosyltransferase activity in sporozoites.**
**a** Salivary gland sporozoite count per mosquito 17 days post-bloodmeal. Data on right includes the complemented parasite line. Data are mean ± s.e.m. from $n = 3$ independent experiments each. **b** Immunofluorescence microscopy of NF54, ΔDPY19 and ΔDPY19 Complemented salivary gland sporozoites with antibodies to CSP (magenta), 5G12 that recognized the EW(Man) epitope (green) and nuclei stained with DAPI. Scale, 5 μm. **c** Western blot analysis of salivary gland sporozoites and uninfected salivary gland lysates (Ui SG) with antibodies to 5G12 that recognize the EW(Man) epitope, as well as TRAP and CSP. The same blot was probed consecutively with each antibody. The C-mannosylated protein in sporozoites (arrow)

and uninfected salivary glands (asterisk) are indicated. Shown is a representative blot from two independent experiments showing the same result. **d, e** Western blot analysis of salivary gland sporozoites with antibodies to 5G12, which recognize the EW(Man) epitope, TRAP and CSP. The same blot was probed consecutively with each antibody. The C-mannosylated protein in NF54 (arrow) is absent from ΔDPY19 sporozoites and the cross-reactive band from salivary glands (asterisk) are indicated. The left blot shows NF54 versus ΔDPY19 c1 and the right blot shows NF54 versus ΔDPY19 c2, both showing the same result. The number of sporozoites included in each lane are indicated. The same blot in **d, e** was probed consecutively with each antibody. Data are representative of $n = 1$ experiment in each of **d, e**.

proteins like TRAP is sufficient to compensate for the absence of tryptophan C-mannosylation. We successfully generated double mutants lacking both the O- and C-glycosylation pathways in asexual stages, which were inhibited for transmission to the mosquito. This defect prevented robust production of sporozoites for phenotyping. It is plausible that loss of both O- and C-glycosylation pathways causes even more severe phenotypes in sporozoites than loss of O-fuccosylation alone[36], given the essential roles played by TSR proteins in these parasites[23–29]. Nevertheless, our study unequivocally demonstrates that DPY19 activity is important for the function of

some, but not all, parasite TSR proteins for transmission to mosquitoes.

## Methods

### Parasite maintenance
*P. falciparum* NF54 (kindly provided by the Walter Reid Army Institute of Research) asexual stages were cultured in human O-positive erythrocytes (Melbourne Red Cross) at 4% haematocrit in RPMI-HEPES supplemented with 0.2% NaHCO$_3$, 7% heat-inactivated human serum (Melbourne Red Cross), and 3% AlbuMAX (ThermoFisher Scientific).

Parasites were maintained at 37 °C in 94% N, 5% CO$_2$ and 1% O$_2$. Gametocytes were cultured in RPMI-HEPES supplemented with 0.2% NaHCO$_3$, and 10% heat-inactivated human serum (Melbourne Red Cross). Gametocytes for transmission to mosquitoes were generated using the "crash" method[62] using daily media changes.

### Transgenic parasites

Highly synchronized schizonts were enriched by using magnet-activated cell sorting (MACS) magnetic separation columns (Miltenyi biotec) and then incubated with the protease inhibitor E64 (Sigma) until parasitophorous vacuole (PV) enclosed merozoite structures (PEMS) could be visualized[63]. Transfections were performed using Amaxa Basic Parasite Nucleofector Kit 2 (Lonza). Purified plasmid DNA (100 μg, Life Technologies) from each plasmid was transfected into *P. falciparum* NF54 and stable transfectants selected[64]. Integration of plasmid DNA constructs was confirmed by Southern blot analysis using the Roche digoxigenin (DIG) system according to the manufacturer's instructions.

To HA-tag the C-terminus of DPY19, homologous target sequences corresponding to the 3'end of *DPY19* was amplified from *P. falciparum* NF54 gDNA using the primers DPY19HA_F 5'-TCGC GGCCGCTAACGTAACAGACTTAGGAGGAGATCCTATAGGTGTTCATC ATCAGGTTC-3' and DPY19HA_R 5'-GTATGCTGCAGCATTGTATAT ATAAAAAAATAATAAATAATCTAAAAAACG-3'. The amplicon was cloned into p1.2-SHA[65] via NotI/PstI.

The pCas9-PfDPY19 plasmid was generated using a modified pUF-Cas9 vector that encodes both Cas9 and single guide RNA (sgRNA)[66,67]. The BtgZI adaptor sequence was replaced with the DPY19 guide DNA sequence 5'-TGCCTCTTTTCTAGTTTAGTTGG-3'[66].

The pΔPfDPY19 plasmid was generated by using pCC4[68]. The homologous target sequence corresponding to the 5' region upstream of the cas9 cut site was amplified using the primers PfDPY19KO_5'_F 5'-GTTATCTCGAGCCAACAATCAACATTCTAAGA CTTG-3' and PfDPY19KO_5'_R 5'-GTATGCTGCAGCCCTAGAAAA-GAGGCATATAAGAGGTAAAGC-3'. The 5' flank amplicon was cloned into pCC4 via XhoI/SpeI. The homologous target sequence corresponding to the 3' region downstream of the Cas9 cut site was amplified using the primers PfDPY19KO_3'_F 5'-GCCAT-GAATTCCTCCTCATTTTCTGGATGACC-3' and PfDPY19KO_3'_R 5'-TGTAACCTAGGCTTGAACCCTCTTTCTCAAGTTAC-3'. The 3' flank amplicon was cloned into pCC4 via EcoRI/AvrII. The homologous flanks in this construct correspond to the upstream and downstream regions of the protospacer-adjacent motif (PAM), which facilitates homologous-direct repair (HDR) resulting in the disruption of the *DPY19* locus.

To construct the DPY19 complementation plasmids, a new pCas9-PfDPY19_Comp plasmid was generated based on an earlier design[69] using a modified pUF-Cas9 vector that encodes both Cas9 and single guide RNA (sgRNA)[66,67]. The BtgZI adaptor sequence was replaced with the DPY19 complement guide DNA sequence 5'-TATTGTAAA-CAACTCTTAAAAGG-3'[66]. The complement repair template (synthesised by GeneArt) comprised of 648 bp of complimentary sequence upstream of the guide sequence and the remainder of the *DPY19* gene sequence was codon-optimised and cloned using SalI/KpnI into a plasmid containing PbEF1alpha promoter, blasticidin resistance gene, camodulin terminator and camodulin promoter (3' flank target sequence). The blasticidin resistance gene was replaced with human *Dihydrofolate reductase* gene (synthesised by GeneArt) and cloned using SwaI/SpeI restriction sites, generating pDPY19Comp.

*P. falciparum* NF54 parasites expressing CTRP-GFP were generated as follows. To drive expression, the 5' UTR of the TSR gene *PTRAMP* (PF3D7_1218000) was amplified from *P. falciparum* NF54 gDNA with the primers atcgcggccgcGCGCAAGAGGAAAAACAATA-TATTC and gatctcgagCTTTAAAAAAAAATAACAAAAGAAAAAAC containing NotI or XhoI sites, respectively, which were used to clone the promoter into pGlux.1[48]. DNA encoding the ER-type signal sequence from PTEX150 (PF3D7_1436300) and the seven tandem TSR domains plus cytoplasmic tail of CTRP (PF3D7_0315200) (synthesised by IDT) comprising of 1717bp was codon-optimised and cloned using XhoI/XmaI into pGlux. Purified plasmid DNA (100μg, Life Technologies) was transfected into *P. falciparum* NF54 and stable transfectants were selected[48].

### Blood stage growth assay

Highly synchronized trophozoite stage parasites were diluted to 0.5% parasitemia at 1% hematocrit. Starting parasitemia was confirmed by flow cytometry (FACSCalibur; BD) using ethidium bromide staining (1:1,000 dilution in PBS). Final parasitemia was determined 48 hours later by FACS as above. For each line, triplicate samples of 50,000 cells were counted in each of the three independent experiments. Growth was expressed as a percentage of NF54.

### SPATR antiserum

Rabbit antisera was raised against PF3D7_0212600 (SPATR) to amino acid residues 60-250 (generated by Genscript). Briefly, the following protein sequence was expressed recombinantly in Escherichia coli as a 6His fusion: GDTCVIFSSSEGNSRNCWCPRGYILCSEEDVLDVQGKL-NEIKNKHERSLVTPLWMKRLCDNSNDVGFKSMSVVIDYELAVLCKDGS NKDYADFEIIGASGYITGEEMIEEQKRNPWYVPRKCTVNNFYLCRKVEN DNVNCSYTPWSDWSACKNNTQKRYRKVRRSNQNNENFCLWNDKIV PRNIMEQTRSC. Purified protein was used to immunize two rabbits and rabbit serum was affinity purified using the immunogen.

### Immunoblotting

Synchronized late asexual schizont cultures were passed over CS magnetic columns (Miltenyi Biotech) to enrich schizont-infected erythrocytes. The 13-TM protein DPY19-HA was extracted from parasite membranes by incubating the schizont saponin pellet in DNAse I at 37 °C for 30 min, resuspended in equal volume 2x Laemmli's buffer and further incubation at 37 °C for 45 min to gently denature the sample for SDS-PAGE. Stage V gametocytes were prepared using the "crash" method and sporozoites were dissected from mosquito salivary glands 17 days post bloodmeal, both were resuspende in 4x Laemmli's buffer for SDS-PAGE. Proteins were separated on NuPAGE 4-12% Bis-Tris polyacrylamide gels with MOPS Buffer (Invitrogen), transferred to nitrocellulose membrane and blocked in 5% (w/v) skim milk-phosphate-buffered saline (PBS)− 0.1% Tween 20 and probed with primary antibodies: rat anti-HA 1:500 (Roche 3F10), rabbit anti-SPATR 1:500 (R3855, Genscript), rat anti-PTRAMP 1:250[38], rabbit anti-MTRAP 1:500[70], rabbit anti-Aldolase (1:4000)[71], mouse anti-5G12[8] (1:1000), mouse anti-HSP70[67] (1:500), rabbit anti-TRAP (1:2000)[72], mouse anti-CSP 2A10[73] (1:9000) followed by HRP-conjugated secondary antibodies (1:1000 (mouse) (Merck), or 1:4000 (rabbit) (Merck), or 1:1000 (rat) (Merck) and the signal was detected using SuperSignal West Pico Chemiluminescent solution (Thermo Scientific). For yeast immunoblots, pure cleaved MTRAP (1000 ng) from GS115 and *DPY19 +* strains and SPATR (1000 ng) from dpy19+ strain were loaded and separated using a Bolt Bis-Tris Plus SDS−PAGE gel. Proteins were transferred onto a nitrocellulose membrane using the iBlot™ 2 Gel Transfer Device (Thermofisher). The membrane was blocked using PBS + 5% skim milk for 1 h at room temperature and was washed three times with PBS + 0.1% tween 20. The membrane was incubated with 10 ml of PBS + 3% skim containing 1 μg/ml of 5G12 as the primary antibody for 1 h at room temperature. The membrane was then washed three times with PBS + 0.1% tween 20. The membrane was then incubated with 10 ml of PBS + 3% skim milk containing Goat Anti-Mouse IgG H&L (HRP) (ab97023) (1:10,000) as the secondary antibody for 1 h at room temperature. The membrane was then washed three times with PBS + 0.1% tween 20 and then incubated with Immobilon ECL (enhanced chemiluminescence) HRP substrate and view on a Chemidoc system.

## Prediction of new domains in *P. falciparum* TSR proteins

AlphaFold 2[74] models for *P. falciparum* proteins with a TSR domain were downloaded from the UniProt database (Q8I2A0, P19597, P16893, C6KT06, C0H4X0, Q8IL45, O97267, O96207, Q8I5M8, Q8IJB7) and the boundaries of predicted globular domains manually assigned. Hitherto unrecognised domains were then classified using structural homology searches with DALI[75], and those with Z scores >2 were annotated in Fig. 1a and Supplementary Data 1.

## Yeast expression and protein purification

A synthetic codon-optimised gBlock containing the TSR domains of MTRAP and SPATR was synthesised by IDT and this was then amplified and cloned into the pPIC9k vector using EcoRI and NotI. Plasmids were sequenced using sanger sequencing. 20 µg of DNA was linearised with SalI and cleaned up using ethanol precipitation. These plasmids were integrated into *P. pastoris* GS115 and *DPY19*+ strains according to the manufacturer's instructions. For protein expression, 10 ml of YPD was inoculated with the protein expression strain and cultured at 30 °C for 24 h at 220 rpm which was then used to inoculate 1 L of BMGY and 30 °C for 24 h at 220 rpm. When OD600 reached 6-8, cells were harvested (1500 g, 20 min, 4 °C) and then resuspended in BMMY medium and cultured at 30 °C for 24 h at 220 rpm. After 24 hours, the supernatant was harvested (9000 g, 30 min, 4 °C). The supernatant was filtered (0.2 µm), a protease inhibitor cocktail tablet was added along with 2 mM EDTA and 0.02% sodium azide. The supernatant was concentrated to 50 ml using an Amicon stirred cell (Millipore; 10-kDa nominal molecular weight limit (NMWL)) and dialyzed (10-kDa NMWL) twice against 2000 ml of binding buffer (50 mM Tris-HCl pH 7.5, 500 mM NaCl, 5 mM Imidazole). The dialyzed sample was filtered (0.2 µm) and was then loaded on a His GraviTrap TALON column. The column was washed with three column volumes of binding buffer, and the protein was eluted using elution buffer (50 mM Tris-HCl pH 7.5, 500 mM NaCl, 500 mM Imidazole). Fractions containing protein, as determined by SDS–PAGE, were pooled. Pooled protein was then passed through a Superdex 75 increase 10/300 GL; GE Healthcare) using 50 mM Tris-HCl and 150 mM NaCl, pH 7.5, as buffer. Factor X Activated (Xa) was used to cleave tags off from the proteins and the tags were captured by passing through His GraviTrap TALON column and collected flowthrough which contained the cleave tag-less protein. Protein was then concentrated using Centricon centrifugal filters (Millipore; 3-kDa NMWL) and the protein was further purified using a Superdex 75 increase 10/300 GL; GE Healthcare) using 50 mM Tris-HCl and 150 mM NaCl, pH 7.5 as buffer. Pure fractions, as determined by SDS–PAGE, were pooled and flash-frozen and stored–at − 80 °C until further use.

## MTRAP-HA immunoprecipitation

Synchronized late trophozoite/early schizont cultures precultured with aspartic protease inhibitor WM382 at 2.5 nM final concentration to reduce MTRAP processing[40] were passed over CS magnetic columns (Miltenyi Biotech) to remove uninfected erythrocytes. Schizont-infected erythrocytes were eluted from magnetic columns with complete RPMI 1640 medium containing WM382. Eluted parasites were pelleted and snap frozen before immunoprecipitation with anti-HA agarose as previously described[8]. Briefly, 1 g pellet of parasites was resuspended in 20 ml of lysis buffer (50 mM Tris-HCl @ pH 7.5, 150 mM NaCl, 0.2% SDS, 2% TX-100, 1 mM EDTA, 50 nM L-321, 1 protease inhibitor tablet) and sonicated on ice for 5 min (30% amplitude, 15 on, 15 off). Proteins were solubilized by rotating at 4 °C for 2 h, centrifuged at 17,000 g 20 min, 4 °C supernatant incubated with 50 µL anti-HA Affinity Matrix (11815016001, Roche) at 4 °C for 2 h while rotating. Beads were collected in spin cups, washed five times in 50 mM Tris-HCl @ pH 7.5, 150 mM NaCl, 0.2% TX-100, 10 nM WM382 and another five times with buffer without 0.2% TX-100 and protein was eluted in 100 mM glycine @ pH 2.6, 200 mM NaCl, 25 nM WM382 before neutralisation in

1 M Tris-HCl pH 9.0. The protein sample was concentrated and snap-frozen and stored −80 °C until further use.

## Intact protein MS analysis

Intact analysis was performed on a Maxis II ETD UHR-QqTOF mass spectrometer (Bruker Daltonics, Bremen, Germany) equipped with a Captivespray source and NanoBooster infusing acetonitrile at 0.2 ml/min to enhance MS signal intensity. Affinity purified protein samples were acidified with 2% formic acid and loaded onto a ProSwift RP-4H 250 mm by 100µm column (Thermo Scientific, San Jose, USA) using buffer A (3% acetonitrile, 0.1% formic acid) for 10 mins and then eluted with a linear 35 min gradient from 3 to 85% buffer B (99.9% acetonitrile, 0.1% formic acid) at a flow rate of 1 µl/min. MS1 Mass spectra were acquired at 1 Hz between a mass range of 150–2200 m/z. Intact mass masses were determined by MaxEnt deconvoluted within DataAnalysis 4.3 (Bruker).

## S-trap protein digestions of rMTRAP and rSPATR

Affinity purified samples were adjusted to 4% SDS, 100 mM TEAB pH 7.1, 10 mM DTT in a volume of 50 µl and then boiled for 10 minutes at 95 °C. Samples were then alkylated with iodoacetamide (50 mM final concentration) for 1 hr in the dark at room temperature. Reduced/alkylated samples were then cleaned up using S-trap micro columns (ProtiFi, USA) according to the manufacturer's instructions followed by digestion with 3 µg of SOLu-trypsin in 100 mM TEAB pH 8.5 (Sigma, 1:30 protease/protein ratio) for 4 hours at 47 °C. Digested samples were collected from the S-traps according to the manufactures instructions and dried down by vacuum centrifugation.

## Analysis of digested rMTRAP and rSPATR using LC-MS

Dried digests were resuspended in buffer A*(0.1% TFA, 2% ACN) and separated using a two-column chromatography set up on a Dionex Ultimate 3000 UPLC (Thermo Fisher Scientific) composed of a Pep-Map100 C18 20 mm × 75 µm trap and a PepMap C18 500 mm × 75 µm analytical column (Thermo Fisher Scientific). The samples were concentrated onto the trap column at 5 µl/min using 0.1% formic acid (FA) for 6 min and then infused into an Orbitrap Fusion™ Eclipse™ Tribrid™ Mass Spectrometer (Thermo Fisher Scientific) at 300 nl/min via the analytical column. Separation on the analytical column was undertaken using buffer A (2% DMSO, 0.1% FA) and buffer B (78% ACN, 2% DMSO and 0.1% FA) by altering the buffer composition from 3% buffer B to 23% buffer B over 29 min, 23% buffer B to 40% buffer B over 10 min, 40% buffer B to 80% buffer B over 5 min, then the composition was held at 80% buffer B for 5 min, and then dropped to 3% buffer B over 1 min and held at 3% buffer B for another 9 min. The Eclipse™ Mass Spectrometer was operated in a data-dependent mode switching between the acquisition of a single Orbitrap MS scan (120k resolution) every 3 seconds followed by Orbitrap HCD scans (maximum fill time 100 ms, AGC 2 × 105 with a resolution of 50k for the Orbitrap MS-MS scans using stepped NCE with 25, 30, 40).

## Analysis of data generated for digested rMTRAP and rSPATR

Identification of C-glycosylated peptides was undertaken using Max-Quant (v1.6.17.0)[76]. The predicted amino acid sequences for rMTRAP and rSPATR were combined into a database and searched, allowing carbamidomethylation of cysteine residues as a fixed modification and the variable modifications of oxidation of methionine and C-glycosylation (C6H10O5; 162.052824 Da on tryptophan, allowing the neutral loss of C4H8O4; 120.04225 Da). Searches were performed with either Trypsin cleavage specificity allowing 2 miscleavage events with a maximum false discovery rate (FDR) of 1.0 % set for protein and peptide identifications. The resulting MS2 scans corresponding to glycosylation events of interest were manually extracted using the Freestyle Viewer (1.7 SP1, Thermo Fisher Scientific) and then visualized to conf26ocalizationtion of the C-glycosylation events using the

Interactive Peptide Spectral Annotator http://www.interactivepeptide spectralannotator.com/PeptideAnnotator.html[77]. To quantify the relative level of C-glycosylation within samples extracted ion chromatograms of the most abundant, as determined by the observed ion intensity, charge states for a given peptide sequence were extracted, ±10 ppm, using the Freestyle Viewer. Peaks were processed with a 11-point Gaussian smooth and the area under the curve calculated. The resulting areas were used to compare relative occupation rates of peptide sequences.

## Thermal denaturation experiments

NanoDSF studies were performed on a Prometheus NT.48 (Nano-Temper). Data recording and initial analysis was performed with PR.ThermControl software. All protein samples were at 5 mg.ml-1 in 50 mM Tris.HCl, 200 mM NaCl @ pH 7.5 using 15 µl of sample per capillary. Experiments were performed in duplicates with the temperature ramp from 15 °C to 95 °C, at 1 °C.min-1 with 20% excitation power.

## Gametocyte preparation and analyses

*P. falciparum* gametocytes were generated using the "crash" method[62] using daily media changes as previously described[78]. Exflagellation was assessed by removing 200 µl of gametocyte culture, briefly centrifuging the culture and resuspending the cell pellet in 10 µl ookinete medium[79,80]. Samples were incubated at room temperature for 15 min to induce gametogenesis and after ~15 min exflagellation was observed by phase contrast microscopy at a magnification of 40x in a Neubauer haemocytometer cell counting chamber. The number of exflagellation centers were enumerated in the 4 × 4 outer grids for each parasite line. For confocal imaging, parasites were fixed and processed as described[81]. For lattice light-sheet microscopy, stage V gametocytes were maintained at 37 °C in all subsequently described steps until activation and imaging. Gametocytes were purified on a 70% Percoll gradient and resuspended in RPMI-HEPES without human serum. Purified gametocytes were labelled with 1.5 µM Di-4-ANEPPDHQ[56], SPY650-tubulin (1:1000) in phenol red-free RPMI-HEPES supplemented with 0.2% NaHCO3. Gametocytes were added to the pre-warmed staining solution and incubated in a water bath at 37 °C. After 1 h, stained gametocytes were washed and resuspended in media containing 10% human serum and stored at 37 °C until imaging. Just prior to imaging, 50-100 µL of the cell suspension (still in medium with serum) was added to 200 µL of ookinete medium that has been supplemented with SPY650-tubulin to enable further uptake of dye molecules during gametogenesis, and maintained at room temperature, mounted on iBidi glass 8-well plates and imaged immediately.

## Confocal immunofluorescence microscopy

Asexual stages, gametocytes and ookinetes were fixed in 4% paraformaldehyde (ThermoFisher Scientific)/0.0075% glutaraldehyde (Merck) in PBS. Immunofluorescence assay performed on exflagellating male gametocytes comparing ΔPfDPY19 and NF54 parasites were fixed in 4% paraformaldehyde (ThermoFisher Scientific)/0.0075% glutaraldehyde (Merck) in microtubule stabilization buffer[82]. All fixed parasites were permeabilized in 0.1% Triton X-100/PBS and probed with rat anti-HA (1:500; Roche 3F10). Asexual stages, gametocytes and ookinetes were fixed in 4% paraformaldehyde (ThermoFisher Scientific)/0.0075% glutaraldehyde (Merck) in PBS. Immunofluorescence assay performed on exflagellating male gametocytes comparing ΔPfDPY19 and NF54 parasites were fixed in 4% paraformaldehyde (ThermoFisher Scientific)/0.0075% glutaraldehyde (Merck) in microtubule stabilization buffer[82]. All fixed parasites were permeabilized in 0.1% Triton X-100/PBS and probed with rat anti-HA (1:500; Roche 3F10)[50], rabbit anti-plasmepsin V (1:1000, R1245)[37], mouse anti-Tubulin clone DM1A (1:300, Sigma), mouse anti-Pfs25 4B7[83] (1:500), mouse anti-CSP clone 2A10[73] (1:2000) and mouse anti-EW(Man) 5G12[8] (1:500).

When necessary, mouse anti-CSP conjugated to AlexaFluor 647 (1:2000) was used. Secondary antibodies were goat anti-rabbit or anti-mouse AlexaFluor 594 and anti-mouse or rat AlexaFluor 488 (1:1000; Invitrogen) in 3%BSA/PBS. Samples were washed in PBS before washing three times in PBS, staining with 2 µg/ml DAPI, and being mounted on coverslips coated with 1% polyethyleneimine beneath cover glasses with Vectashield (Vector Labs). Images were acquired using an LSM880 confocal microscope using a ×60 objective (Carl Zeiss MicroImaging) and ZEN black v14.0 software (Zeiss). Images were analyzed using FIJI v1.0 (ImageJ), Adobe Photoshop 2021 v22.4.1 and assembled using Adobe Illustrator 2021 v25.4.

## Lattice light-sheet microscopy

Time lapse live cell data was acquired using a Lattice Light-Sheet 7 (L−S7 - Zeiss − Pre-serial). Light-sheets (488 nm and 633 nm) of length 30 µm with a thickness of 1 µm were created at the sample plane via a 13.3 x, 0.44 NA objective. Fluorescence emission was collected via a 44.83 X, 1 NA detection objective using a pco.edge 4.2 (PCO) camera. Aberration correction was set to a value of 182 to minimise aberrations as determined by imaging the Point Spread Function using 100 nm fluorescent microspheres at the coverslip of a glass bottom chamber slide. Resolution was determined to be 454 nm (lateral) and 782 nm (axial). Data was collected with a range of frame rates of 5 ms and a y-step interval of 300 nm. Imaging commenced immediately upon addition of 50-100 µL of either NF54 or DPY19 C2 parasites at a volume rate of one volume (250 × 250 × 30 µm) per second. Light was collected via a multi-band stop, LBF 405/488/561/633, filter. Data was subsequently deskewed then deconvolved using a constrained iterative algorithm and 20 iterations in the ZEN software (Zeiss). Images were collected at 23 °C. All images were rendered in 3D using IMARIS (Bitplane) for visualisation and manual analysis. All events were assessed manually for timings relating to full, partial or no egress for microgametocyes and macrogametocytes.

## Mosquito infection and analysis of parasite development

One- to three-day old female *Anopheles stephensi* mosquitoes (kindly provided by M. Jacobs-Lorena, Johns Hopkins University) were fed a bloodmeal of asynchronous gametocytes 0.3% stage V gametocytemia, via water-jacketed glass membrane feeders. Mosquitoes were sugar starved for two days post-bloodmeal to enhance the population for bloodfed mosquitoes. Surviving mosquitoes were provided sugar cubes and water wicks ad libitum. At 7 days post-bloodmeal, midguts were dissected from cold-anesthetized and ethanol killed mosquitoes and stained with 0.1% mercurochrome, and oocysts per mosquito were counted by microscopy.

## Preparation of ookinetes

At 23 h post-bloodmeal, infected midguts were dissected, and contents pooled from 20 mosquitoes (per group) to isolate ookinetes as previously described[84]. Ookinete time courses were performed by dissecting infected midguts at 23 h, 29 h and 32 h post-bloodmeal. Briefly, midguts were torn and punctured, and a pipet was used to facilitate the release of the blood meal into tube containing ~200 µl PBS on ice. Midguts were pelleted and 1 mL of 0.15% saponin/PBS was used to resuspend the midgut pellet containing ookinetes and incubated on ice for 10 min. Pellets were washed thrice with cold PBS, resuspended in 50 µl PBS and 5 µl was spotted on a microscope slide and Giemsa stained. Ookinetes were quantified by counting the number of stage I-V ookinetes (round, knob, stem, enlarged stem, mature)[44] for 25 minutes per slide by light microscopy of blinded samples.

## *An. stephensi* immune-responsive gene expression

Infected midguts were dissected 27 h post-bloodmeal from mosquitoes to check for the presence of ookinetes, SRPN6 and LL3[36]. Briefly, mosquitoes were cold-anesthetized and ethanol killed. Midguts from

25 mosquitoes per group were dissected on ice-cold PBS and frozen immediately on dry ice. RNA was purified using TRI Reagent (Sigma) and complementary DNA (cDNA) was prepared using a SensiFast cDNA synthesis kit (Bioline) according to the manufactures' instructions. qRT-PCR was performed using a LightCycler 480 (Roche) using oligonucleotides that recognize *P. falciparum Pfs25* (5′-GAAA TCCCGTTTCATACGCTTG-3′ and 5′-AGTTTTAACAGGATTGCTTGTA TCTAA-3′), *CTRP* (5′-GAATGGAGTCCCTGTCCTGA-3′ and 5′-TGGT CCTTTCCTTTCCCTTT-3′), *18S* (5′-TATTGCTTTTGAGAGGTTT TGTTACTTTG-3′, and 5′- ACCTCTGACATCTGAATACGAATGC-3′), and *An. stephensi LL3* (5′-CTCACAGACACCATCGATAGC-3′ and 5′-GATTCGTGCTTCACTTTCGTG-3′) and *rps7* (5′-TGCGGTTTCA-GATCCGAGTTC −3′ 5′- TTCGTTGTGAACCCAAATAAAAATC −3′). SRPN6 oligonucleotides were reported previously[84].

### Hepatocyte culturing and infection
HC-04 hepatocytes[85] were maintained in Iscove's Modified Dulbecco's Medium (IMDM, Life Technologies 12200) containing 1x penicillin and streptomycin and supplemented with 5 % of heat-inactivated fetal bovine serum and passaged every 2-3 days using trypsin. The day prior infections, cells were resuspended in a single cell solution and 100,000 cells per well were seeded in a 96-well plate. Salivary glands were dissected in Schneider media and sporozoites were extracted as detailed above. Dissection time was kept under 1 h to ensure sporozoites' viability, limiting us to 50 mosquitoes per line. Parasites were pelleted at 10,000 x g for 3 min, resuspended in IMDM supplemented with 10 % human serum and 0.5 mg/mL dextran-FITC, and spun onto hepatocytes at 100 x g for 3 min. After 4 h incubation at 37 °C, cells were trypsinized, half of them were re-plated while the other half was washed in DPBS twice and dextran-FITC positivity was analysed by flow cytometry (BD LSRFortessa™ X-20). Twenty-four hours post sporozoite addition, cells were fixed and permeabilised using the BD Cytofix/Cytoperm™ kit (BD Bioscience 554714). α-PfCSP 2A10 monoclonal antibody conjugated to AlexaFluor 647 was used at 1 µg/mL to stain for sporozoites inside HC-04 cells and the percentage of intracellular parasites was analysed by flow cytometry using FlowJo for Mac v10.8 software.

### Statistics
Statistical analyses were performed using the Kruskal–Wallis one-way ANOVA to compare two mutant clones to NF54 throughout this study and the Mann–Whitney test was used to compare one mutant clone to NF54. Oocysts were compared using Kruskal-Wallis test with Dunn's correction whilst gamete egress and mosquito infection prevalence was compared using the chi-square test. Analyses were performed using Graphpad Prism 9 for macOS. $P < 0.05$ was considered statistically significant.

### Reporting summary
Further information on research design is available in the Nature Research Reporting Summary linked to this article.

## Data availability
The biological data generated in this study are provided in the Supplementary Information. Biological tools produced in this study are available from the corresponding authors upon request. Mass spectrometry proteomics data have been deposited to the ProteomeXchange Consortium via the PRIDE partner repository with the dataset identifier PXD033470. Source data are provided with this paper.

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

## Acknowledgements

We thank the Melbourne Red Cross for erythrocytes, the US Naval Medical Research Centre for HC-04 cells, the Melbourne Mass Spectrometry and Proteomics Facility of The Bio21 Molecular Science and Biotechnology Institute for access to MS instrumentation, Jake Baum for MTRAP antibodies, Chetan Chitnis for PTRAMP antibodies, Alan Cowman for TRAP and Aldolase antibodies and Fidel Zavala for CSP antibodies. Monoclonal Antibody 4B7 anti-*P. falciparum* 25 kDa Gamete Surface Protein (Pfs25), MRA-28, was obtained through BEI Resources NIAID, NIH, contributed by David C. Kaslow. We acknowledge Sabrina Caiazzo, Lachlan Whitehead, Julie Healer and Melissa Hobbs for technical assistance. This work was supported by the Australian National Health and Medical Research Council (GNT1139546 to E.D.G.-B. and J.A.B. and GNT1123727 to J.A.B.) and an Australian Research Council Discovery Project Grant (210100362 to N.E.S.). We also acknowledge Victorian State Government Operational Infrastructure Support and Australian Government NHMRC IRIISS. N.E.S. is supported by an Australian Research Council Future Fellowship (200100270), E.D.G.-B. is supported by Brian M. Davis Fellowship and J.A.B. is supported by an Australian National Health and Medical Research Council L1 Leadership Fellowship (1176955).

## Author contributions

Experiments were performed by S.L., R.M., A.J., N.G., S.D.M., L.V., C.E., R.W.J.S., N.E.S. and J.A.B. Methodology was performed by S.L., N.G., C.E., M.T.O., N.E.S., K.L.R., E.D.G.-B. and J.A.B. Data was analysed by S.L., R.M., A.J., N.G., R.W.J.S., K.L.R., N.M.S., N.E.S., K.L.R., E.D.G.-B. and J.A.B. Data was interpreted by N.G., K.L.R., E.D.G.-B. and J.A.B. Experiments were conceived by E.D.G.-B. and J.A.B. All authors contributed to preparing this manuscript.

## Competing interests

The authors declare no competing interests.

## Ethics statement

All experimental protocols involving the HC-04 human hepatocyte cell line were reviewed and approved by the Walter and Eliza Hall Institute of Medical Research Biosafety Committee.
