## [Peer Review File · Nature Communications]

Reviewer comments, first round review: –

Reviewer #1 (Remarks to the Author):

Presented here is the first investigation of the role of the *P. falciparum* Dpy19 – a C-mannosyltransferase that glycosylates TSR domains – in transmission stages of the human malaria parasite *P. falciparum*. The authors definitively show that PfDpy19 is essential for transmission of *P. falciparum* by demonstrating that the enzyme plays a role in egress of gametes in the mosquito midgut, and that it is essential for ookinetes to successfully invade the mosquito midgut epithelium and form oocysts. To provide a mechanism for this phenotype, the authors demonstrate the glycosyltransferase activity of PfDpy19 *in vivo* and use a variety of approaches to show that preventing C-mannosylation of key TSR-containing proteins may destabilize the proteins and hinder their secretion.

The manuscript is well-written, the experiments are well-conceived and well-executed, and the hypotheses are well-supported by the presented data. This reviewer recommends publication with only minor revisions.

Minor issues:

Line 69. "C-glycosylation has been detected for TRAP in *P. falciparum* (31) and *P. vivax* (32) sporozoites."

Reference 32 (Swearingen 2017) showed that unlike in *P. falciparum*, *P. vivax* TRAP is *not* C-mannosylated *in vivo*. Song 2012 (PMID: 23236185) also failed to observe C-mannosylation of PvTRAP that was recombinantly expressed in HEK293T. It has also been shown that PyTRAP and PyCSP are C-mannosylated (Swearingen 2019; PMID: 30523691).

Line 100. "The single enzyme likely to perform this function, *P. falciparum* DPY19 (PF3D7_0806200), shares 36.8% similarity and 13.1% identity with the well-studied *Caenorhabditis elegans* DPY19 (CCD62139.1) (36).

Reference 36 (Chen et al 2012) does not mention Dpy19, but reference 31 (Swearingen 2016) identified PF3D7_0806200 as a putative homolog of Dpy19.

Reviewer #2 (Remarks to the Author):

The work by Boddey and colleagues investigates the importance of the C-mannosyltransferase DYP19 for the life cycle progression of the malaria parasite *Plasmodium falciparum*. They investigate the subcellular localization, the effect of deletion on the stability of TSR domain containing proteins, and the importance of DYP19 for transmission.

While the authors attempt a comprehensive analysis of DYP19 and provide a very large amount of data, these data and the corresponding text are often confusing, sometimes contradicting. Thus, in the current form of the paper, the reader cannot readily access the significance of the work. In addition, several major claims are not well covered by data and, thus, the paper would greatly benefit from a more conservative interpretation of the results ("definitive proof" may have a very short half-life in science).

Below are some specific comments to help with the process of clarification.

The description and reference to Fig. 3a in line 161 is unclear and potentially misleading. While the figure shows a signal for rMTRAP in the CeDPY19-expressing strain, the text says that rSPATR was only obtained when CeDPY19 was co-expressed. And the caption of Fig. 3a states that the detected epitope is not even present in SPATR. This sentence would clearly benefit from revision. A Suppl. Fig. showing the recombinant expression would help here or the statement "data

not shown". Referring to Fig. 3a only makes sense in the context of glycosylation. Also, adding a "kDa" to Fig. 3a would be beneficial. Is not immediately clear we are looking at a Western blot, perhaps changing "5G12" to "anti-EW(Man)" would also help.

Is the statement in line 163 based on Suppl. Fig. 4a or b? If I interpret Suppl. Fig. 4b correctly, then un-, mono-, and diglycosylated rSPATR occur at relatively equal abundance. Please clarify. As these data are important for the conclusion that C-glycosylation is key for protein stability, the paper would benefit from revising Suppl. Fig. 4.

It is unclear how the quantification in Fig. 3c was done. Suppl. Fig. 6 indicates it was the area under the curve. The graph of this read out would clearly strengthen the paper.

From Fig. 3d: It is unclear which are the samples with or without glycosylation. And the color coding is, let's say, suboptimal. Please expand the caption. What does -/+ Trp(Man) mean? What are the replicates? If this information is in the Methods, please provide it here too.

Line 112 to 118: While DPY19 indeed appear to have a perinuclear localization, more (quantitative) imaging data are needed to support the claim that the enzyme is localized to the ER or subdomains thereof.

Paragraph starting in line 139: Up to this point, the paper would benefit from stating that DPY19 deletion leads to less expression of TSR proteins, potentially because the proteins are less stable. While Fig. 3d indeed shows less-stable proteins, here it is "just" less expressed.

Fig. 2d: I assume there is an error and it should be "kDa" and not "Kb". Please check.

Line 166: Are these "data not shown"? If so, please state in the text.

Again, the statement in line 180 would be much stronger if a quantification of the different proteoforms would be provided.

Line 183: If the 5G12 antibody is indeed well-validated, then please provide some references here or edit this sentence.

Please expand the caption of Fig. 3e and explain what the arrows highlight. Knowing what the two bands likely represent would be helpful for the reader. Was this blot also probed consecutively with each antibody as the one shown in Fig. 2d? If so, please state. Also, I think you provide compelling data that MTRAP is C-mannosylated but "unambiguously" (line 188) still seems too strong in the context of Fig. 3e.

Line 197: In contrast to what is written here, Suppl. Fig. 7b shows that PTRAMP is expressed in NF54 stage V gametocytes. Please clarify.

Line 210: Although the microscopy is breath taking (congratulations), it seems highly unlikely that three rounds of genome replication are completed when the "mitotic spindle" became visible at 1 min and 19.8 seconds. Thus, this sentence would benefit from revision.

Fig. 4c and h: "None" in h probably corresponds to "no" in c, perhaps consider harmonizing.

Fig. 4j is described in the text before 4i, this should be changed. The sentence starting in line 233 needs a reference to the data on rounding, or a "data not shown" statement. Also, female macrogametocytes and male microgametocytes are tautologies, perhaps revise.

Fig. 4j and corresponding text: Please explain what "macrogamete egress" is? Or should it be "macrogametocyte egress" ?

The captions of Fig. 5a and b are not in order. How many ookinetes were analyzed per time point and condition? Also, the caption indicated that the \pm s.e.m. is shown but it is not.

Line 261 and further below: It is really irritating that Fig. 6a is discussed before Fig. 5d, 5e, and 5f.

Fig. 5d would benefit from providing more information on what "lower and higher forces of infection" means. Different gametocytemias?

Line 275: In absence of additional data on, e.g., the motility of mutant ookinetes, a more conservative interpretation of the lower LL3 expression is warranted and other explanations should be mentioned.

Fig. 5f: If normalized to aldolase as written on the y-axis, how come that the GFP expression is in both cases 100.0?

The statement in line 320 is unclear and potentially wrong. TRAP mannosylation appears to depend on DPY19, but that does not necessary mean that there are no other C-mannosyltransferases across the life cycle. Please revise.

Line 392: For clarity, should it not be C-mannosylation?

Line 431: The data providing evidence that "DPY19 activity in the early secretory pathway is important" are very scarce and thus statement is too bold.

Reviewer #3 (Remarks to the Author):

The study by Lopaticki is a thorough and well-thought through exploration of C-mannosylation in the malaria parasite, which up until recently was not thought to possess glycosylation processes at all. Building on the work from the same group (having proven O-linked glycosylation) the team now demonstrate that C-mannosylation occurs and, perhaps more importantly, shows strong evidence that it is mediated by a homologue of the C-mannosyltransferase DPY19. The work uses a suite of technologies and appears (to this reviewer) as definitive in its proof that DPY19 in *P. falciparum* is responsible for C-mannosylation of key TSR domain containing proteins that are critical for transmission of the parasite through the mosquito.

The narrative of the discovery work I find clear and generally compelling. My only major issues are where there is a slight tendency to skip over experiments that, on their own, are only partially supportive of the conclusions being stated, but on the whole (together with everything else) do indeed still stand up. For example several key controls (negative controls) I would expect that are lacking. I would expect some of these should be corrected for completeness and to add weight to the overall arguments made.

Detailed comments:

1. Introduction L53: Have the authors looked at *Chromera velia* - a possible evolutionary ancestor to Apicomplexa?
2. Results L97: I am skeptical whether alpha-fold 2 really adds anything here. My inclination would be to drop this as it really doesn't add anything to the paper i.e. speculative. A follow up perhaps but for now since none of this is explored I'd drop.
3. Results L116/F1f: I really feel that given the quality of the lightsheet microscopy this image is weak and doesn't support ER localisation. There is no negative control (to check this isn't a bad batch of anti-HA) and the distribution is not convincing. Is the ring even really a ring or a merozoite? The rest to me looks pretty much speckly i.e. uninformative. I'd suggest better images or if definitive proof of ER is required immuno-EM (though that is also quite hit and miss).
4. Results L137: A question more than a comment - why is DPY19 there if its is dispensable? Could other conditions be explored to show it does serve a blood stage function? Shaking cultures? Nutrient deprivation? This is a recurring problem in asexual KOs, absence of evidence isn't evidence of absence!
5. Results L161: I don't understand how to interpret Figure 3a. The reference to it says rSPATR is discussed in this sentence but the blot ONLY shows rMTRAP as showing up? Is tis miss-labeled or am I mis-interpreting what is being shown/described.

6. Results L180: I would avoid terminology like "remarkable". Perhaps qualify expected and unexpected.
7. Results L189/Figure 3e: I'd really like a negative control here, immuno IPs are tricky and showing the banding pattern looks ok but this can be just heavy and light chains. I'd like to see the same blot without an IP ab (show that there's nothing x-reactive) and also a pre immune serum (or unrelated ab) that shows no x-reactivity. This would make it more convincing and less prone to being a mis-interpreted result.
8. Results L225: The movies are amazing, but there is SO much heterogeneity in exflagellation events to my (admittedly untrained eye) I find Suppl. Movie 4 not entirely dissimilar from WT that I have seen.... i.e. Is this really a mutant phenotype?
9. Results L244: Are the authors not surprised they didn't see the same phenotype as the MTRAP KO - i.e. a super tadpole like bundle of flagellum in unruptured RBC/PVM? I was expecting something similar to this but it seems this is not what was seen? I'd be keen to see this commented on.
10. L307/Figure 6b: There's a danger here that we are looking at N=1 sporozoites for the KO. Is there any way of increasing the numbers shown to really demonstrate the 5G12 staining of KO parasites is actually really (consistently) different?

Reviewer #1 (Remarks to the Author):

Presented here is the first investigation of the role of the *P. falciparum* Dpy19 – a C-mannosyltransferase that glycosylates TSR domains – in transmission stages of the human malaria parasite *P. falciparum*. The authors definitively show that PfDpy19 is essential for transmission of *P. falciparum* by demonstrating that the enzyme plays a role in egress of gametes in the mosquito midgut, and that it is essential for ookinetes to successfully invade the mosquito midgut epithelium and form oocysts. To provide a mechanism for this phenotype, the authors demonstrate the glycosyltransferase activity of PfDpy19 *in vivo* and use a variety of approaches to show that preventing C-mannosylation of key TSR-containing proteins may destabilize the proteins and hinder their secretion.

The manuscript is well-written, the experiments are well-conceived and well-executed, and the hypotheses are well-supported by the presented data. This reviewer recommends publication with only minor revisions.

Minor issues:

Line 69. “C-glycosylation has been detected for TRAP in *P. falciparum* (31) and *P. vivax* (32) sporozoites.”

Reference 32 (Swearingen 2017) showed that unlike in *P. falciparum*, *P. vivax* TRAP is *not* C-mannosylated *in vivo*. Song 2012 (PMID: 23236185) also failed to observe C-mannosylation of PvTRAP that was recombinantly expressed in HEK293T. It has also been shown that PyTRAP and PyCSP are C-mannosylated (Swearingen 2019; PMID: 30523691).

Response: We thank the reviewer for their positive feedback re: this study. We have now altered the text, as follows “C-glycosylation has been detected in sporozoites for TRAP in *P. falciparum*{Swearingen, 2016 #14600} and TRAP and CSP in *P. yoelii*{Swearingen, 2019 #14790}. Sporozoites are the infectious form of the parasite introduced by mosquitoes during blood feeding responsible for initiating mammalian liver infections.”

Line 100. “The single enzyme likely to perform this function, *P. falciparum* DPY19 (PF3D7_0806200), shares 36.8% similarity and 13.1% identity with the well-studied *Caenorhabditis elegans* DPY19 (CCD62139.1) (36).

Reference 36 (Chen et al 2012) does not mention Dpy19, but reference 31 (Swearingen 2016) identified PF3D7_0806200 as a putative homolog of Dpy19.

Response: We have now included the citation after the *Plasmodium* accession number.

Reviewer #2 (Remarks to the Author):

The work by Boddey and colleagues investigates the importance of the C-mannosyltransferase DYP19 for the life cycle progression of the malaria parasite *Plasmodium falciparum*. They investigate the subcellular localization, the effect of deletion on the stability of TSR domain containing proteins, and the importance of DYP19 for transmission.

While the authors attempt a comprehensive analysis of DYP19 and provide a very large amount of data, these data and the corresponding text are often confusing, sometimes contradicting. Thus, in the current form of the paper, the reader cannot readily access the significance of the work. In addition, several major claims are not well covered by data and,

thus, the paper would greatly benefit from a more conservative interpretation of the results (“definitive proof” may have a very short half-life in science).

Below are some specific comments to help with the process of clarification.

The description and reference to Fig. 3a in line 161 is unclear and potentially misleading.

While the figure shows a signal for rMTRAP in the CeDPY19-expressing strain, the text says that rSPATR was only obtained when CeDPY19 was co-expressed. And the caption of Fig. 3a states that the detected epitope is not even present in SPATR. This sentence would clearly benefit from revision. A Suppl. Fig. showing the recombinant expression would help here or the statement “data not shown”. Referring to Fig. 3a only makes sense in the context of glycosylation. Also, adding a “kDa” to Fig. 3a would be beneficial. Is not immediately clear we are looking at a Western blot, perhaps changing “5G12” to “anti-EW(Man)” would also help.

Response: We have repeated the experiment in Fig 3a in a slightly different way to address the reviewers’ concerns and updated the figure, caption and text accordingly. We induced *P. falciparum* TSR protein secretion in six of our yeast lines (no vector, SUMO-MTRAP, SUMO-SPATR: all +/- constitutive co-expression of CeDPY19) at equivalent cell densities. An equal volume of each culture supernatant was used to perform new Western blots (see revised Fig 3a). We first probed the membrane with anti-FLAG to measure expression of each TSR protein: this indicated robust expression of the FLAG-SUMO-MTRAP fusion protein in the presence or absence of CeDPY19. By contrast, very little FLAG-SUMO-SPATR was obtained without co-expression of CeDPY19, likely because the protein requires this modification for stability and/or secretion. The same membrane was then probed with 5G12 (anti-EWMan): this confirmed C-mannosylation of MTRAP when CeDPY19 was co-expressed. Since SPATR possesses a DW(Man) in place of EW(Man) in MTRAP that is recognized by 5G12, rSPATR was not recognized by the antibody as expected, demonstrating specificity of the antibody. Nonetheless, mass spectrometry confirmed that rSPATR was indeed C-mannosylated at the DW(Man) epitope but only when co-expressed with CeDPY19 (Supp Fig 4).

Is the statement in line 163 based on Suppl. Fig. 4a or b? If I interpret Suppl. Fig. 4b correctly, then un-, mono-, and diglycosylated rSPATR occur at relatively equal abundance. Please clarify. As these data are important for the conclusion that C-glycosylation is key for protein stability, the paper would benefit from revising Suppl. Fig. 4.

It is unclear how the quantification in Fig. 3c was done. Suppl. Fig. 6 indicates it was the area under the curve. The graph of this read out would clearly strengthen the paper.

Response: We understand the concerns raised and have modified Supp Fig 4 and 5, as well as their captions, to make our quantitation process clear. In essence, site occupancy was estimated by integrating the relevant peaks in extracted ion chromatograms (EICs).

From Fig. 3d: It is unclear which are the samples with or without glycosylation. And the color coding is, let’s say, suboptimal. Please expand the caption. What does +/- Trp(Man) mean? What are the replicas? If this information is in the Methods, please provide it here too.

Response: We’ve now made some changes to this plot, which hopefully make it easier to interpret. It depicts duplicate melt curves for the rMTRAP samples grown with and without CeDPY19 ie. with and without Trp(Man) (tryptophan mannosylation). The colours for each

curve had been selected from a colour-blind friendly palette to ensure the differences are more clear and we've also used dashed lines to discriminate between glycosylated and unglycosylated samples. We trust these changes make the data clearer, as requested.

Line 112 to 118: While DPY19 indeed appear to have a perinuclear localization, more (quantitative) imaging data are needed to support the claim that the enzyme is localized to the ER or subdomains thereof.

Response: We thank the reviewer for their comments. In metazoans, C-mannosylation is conferred to proteins in the ER (Hoffsteenge et al J Biol Chem 2001). To address this in *P. falciparum*, we have now prepared fresh DPY19-HA parasites and undertaken IFAs with the ER markers ER-calcium binding protein (ERC) and plasmepsin V (PMV). The ERC labelling was unfortunately problematic in these repeat experiments, perhaps due to degradation of the antibody stock, which we do not possess more of. However, IFA with the rabbit anti-PMV antibodies revealed the same perinuclear ER labelling for this protein that we and others have shown previously (Klemba et al Mol Biochem Parasitol 2005; Russo et al Nature 2010; Boddey et al Nature 2010; Sleebis et al PLoS Biology 2014) and the rat anti-HA signal in DPY19-HA was clearly shared with PMV in the cells we observed. To limit any potential issue with previous experiments or confusion about the subcellular localisation, which we feel is not central to the main conclusions of the paper, we have softened the conclusion to "most likely localized to the ER (see below)", removed the figure panels from the earlier version including other lifecycle stages and any reference to subdomains of the ER. We have also edited the results section as follows: "C-mannosylation of metazoan proteins is conferred in the ER⁴. To determine the subcellular localization of DPY19 in *P. falciparum*, transgenic NF54 parasites were produced, whereby the *DPY19* gene was tagged at the C-terminus with a triple hemagglutinin (HA) epitope. Integration of the HA epitope was demonstrated by Southern blot (Supplementary Fig. 2a) and successful tagging confirmed by immunoblot (Fig. 1e). Immunofluorescence microscopy revealed a perinuclear localization for DPY19-HA that co-localized with the ER-resident protease plasmepsin V³⁸ in ring-stage parasites (Fig. 1f). The distribution pattern for DPY19-HA indicates that this enzyme is most likely localized within the ER, similar to metazoans."

Paragraph starting in line 139: Up to this point, the paper would benefit from stating that DPY19 deletion leads to less expression of TSR proteins, potentially because the proteins are less stable. While Fig. 3d indeed shows less-stable proteins, here it is "just" less expressed.

Response: We thank the reviewer for this comment and agree. The text has been modified, and now reads "Immunoblotting and densitometric analysis indicated that levels of SPATR and MTRAP but not PTRAMP were decreased in $\Delta DPY19$ parasites (Fig. 2d). Therefore, DPY19 activity is necessary to maintain normal levels of some but not all TSR proteins in *P. falciparum*. This is potentially because enhanced folding of the C-mannosylated TSR domain stabilizes the protein in parasites."

Fig. 2d: I assume there is an error and it should be "kDa" and not "Kb". Please check.

Response: The reviewer is correct and we have corrected this mistake.

Line 166: Are these "data not shown"? If so, please state in the text.

Response: The data supporting this have now been clarified, please see Fig 3a and SI Fig 4.

Again, the statement in line 180 would be much stronger if a quantification of the different proteoforms would be provided.

Response: The localization and quantification of the various glycoforms is now provided. Please refer to Fig 3c, Supp Fig 5, Supp Fig 6.

Line 183: If the 5G12 antibody is indeed well-validated, then please provide some references here or edit this sentence.

Response: The antibody has been validated in two prior studies (John et al Nature Chemical Biology 2021 and Mao et al JACS 2021). We include both references within the sentence “Immunoblotting these proteins with the monoclonal antibody 5G12, which is specific for tryptophan C-mannosylation and recognises the H/F/L/Q/EW(Man) epitopes^{8,41}, confirmed the C-mannosylation of rMTRAP when co-expressed with CeDPY19 (Fig. 3a).” To avoid confusion, we removed the statement regarding “well-validated”.

Please expand the caption of Fig. 3e and explain what the arrows highlight. Knowing what the two bands likely represent would be helpful for the reader. Was this blot also probed consecutively with each antibody as the one shown in Fig. 2d? If so, please state. Also, I think you provide compelling data that MTRAP is C-mannosylated but “unambiguously” (line 188) still seems too strong in the context of Fig. 3e.

Response: To reduce any confusion, the statement has now been clarified as follows. “HA antibodies specifically identified MTRAP-HA as two protein species, p58 and p35, as previously described⁴². Probing the same membrane with 5G12 antibodies identified that the same bands contained the modified EW(Man) epitope(s), demonstrating that tryptophan C-mannosylation occurs on MTRAP in *P. falciparum* blood stages (Fig. 3e).” The arrows have also been removed to further avoid confusion.

Line 197: In contrast to what is written here, Suppl. Fig. 7b shows that PTRAMP is expressed in NF54 stage V gametocytes. Please clarify.

Response: We thank the reviewer for identifying this mistake. The lanes in Supp. Fig. 7b have now been correctly labelled, as follows: lane 1, NF54 schizonts; lane 2, NF54 stage V; lane 3, Δ DPY19 c2 stage V. The figure legend has also been updated, as follows: “Detection of PTRAMP and SPATR expression by immunoblot in asexual blood stages but not in stage V gametocytes.”

Line 210: Although the microscopy is breath taking (congratulations), it seems highly unlikely that three rounds of genome replication are completed when the “mitotic spindle” became visible at 1 min and 19.8 seconds. Thus, this sentence would benefit from revision.

Response: The sentence has been modified as follows, “Upon addition of xanthuric acid, NF54 microgametocytes shifted from falciform to a round shape indicating activation had occurred, and evidence of mitotic spindle formation was visible by a single punctate region of SPY-tubulin labelling (Fig. 4a).”

Fig. 4c and h: “None” in h probably corresponds to “no” in c, perhaps consider harmonizing.

Response: The term “no” in Fig. 4c is a Δ DPY19-unique phenotype corresponding to ‘no egress’ by the gametocyte BUT successful exflagellation still occurred from the residual body inside the erythrocyte. The term “none” in Fig. 4h corresponds to no exflagellation. To make this clear, we have made the following changes. The term “during microgametogenesis

leading to exflagellation" was added to the legend of Fig 4 a and b; "during microgametogenesis with exflagellation from the residual body inside the erythrocyte" added to the legend of Fig 4c. Further, the key in Fig 4h now states "Full egress", "Partial egress", "No egress" and "Failed (no exflag)".

For females, this may also have been confusing, so we have updated the Fig. 4i legend to state "Quantification of full egress (purple) or failed egress (blue) of macrogametes from the erythrocyte from lattice light-sheet imaging."

We also draw the reviewer's attention to repeated experiments conducted during the review period with the lattice light-sheet microscope in which 27 additional female egress events were recorded and evaluated. 6/9 NF54 females egressed successfully while 5/18 DDPY19 females egressed successfully. Addition of these data to those in Fig. 4i provides sufficient power indicating that the difference is statistically significant ($P=0.0470$, Fisher's exact test). These new data are included in the revision and the results section now reads: "Female gametocytes undertake gametogenesis differently to males ^{42,44}. Imaging with lattice light sheet microscopy revealed that all NF54 and $\Delta DPY19$ macrogametocytes switched from falciform to round after activation however fewer $\Delta DPY19$ macrogametocytes egressed the erythrocyte compared to NF54 (Fig. 4f, g, i, Supplementary Mov. 6, 7) demonstrating that DPY19 is important for egress by females, as well as males."

Fig. 4j is described in the text before 4i, this should be changed. The sentence starting in line 233 needs a reference to the data on rounding, or a "data not shown" statement. Also, female macrogametocytes and male microgametocytes are tautologies, perhaps revise.

Response: The order has now been modified so that the text follows the panels sequentially. "Imaging with lattice light sheet microscopy revealed that all NF54 and $\Delta DPY19$ macrogametocytes switched from falciform to round after activation and shared approximately equal frequencies of successful egress and failed egress by the macrogametocytes (Fig. 4f, g, i, Supplementary Mov. 6, 7). "

We have added the references and removed the tautologies, using "male and female gametes" to avoid confusion.

Fig. 4j and corresponding text: Please explain what "macrogamete egress" is? Or should it be "macrogametocyte egress" ?

Response: PMID 27111866 describes activated gametocytes that leave the red blood cell as 'egressing gametes' (see Fig. 2 of the paper). In the case of females, the time between activation/rounding up and parasite egress from the erythrocyte is considerable (>10 minutes in our study) and we consider it consistent to describe the egressing female parasite a macrogamete. See also the legend for Fig. 4f, g.

The captions of Fig. 5a and b are not in order. How many ookinetes were analyzed per time point and condition? Also, the caption indicated that the \pm s.e.m. is shown but it is not.

Response: We thank the reviewer for identifying the mistake and have corrected the order of the figure legend. The number of ookinetes (n:) for each condition has now been added above each histogram (range 112-298). The caption regarding s.e.m. was in reference to the repeated ookinete quantification experiments included in Supp. Fig. 7e. Owing to the inclusion of additional data in Supp Fig. 7, we have now moved this ookinete quantification from Supp Fig. 7 into Fig. 5 (see b, c).

Line 261 and further below: It is really irritating that Fig. 6a is discussed before Fig. 5d, 5e, and 5f.

Response: The sentence has now been deleted and is described in the next results section on sporozoites.

Fig. 5d would benefit from providing more information on what “lower and higher forces of infection” means. Different gametocytemias?

Response: The terminology was in reference to the clearly different oocyst infection burdens in the midguts between NF54 in the two experiments. These differences are common and may be due to a number of factors. This statement has been clarified the Fig. 5d, e legend.

Line 275: In absence of additional data on, e.g., the motility of mutant ookinetes, a more conservative interpretation of the lower LL3 expression is warranted and other explanations should be mentioned.

Response: We understand the reviewer’s point of view and appreciate it. We have repeated the RT-qPCRs from n=3 independent transmission experiments to quantify mRNA expression in infected and uninfected mosquito midguts for: *P. falciparum* *Pfs25* (gametocytes, zygotes, ookinetes), *CTRP* (ookinetes), *An. stephensi* *LL3* (immune responsive transcription factor upregulated following ookinete invasion) and *SRPN6* (immune responsive protease inhibitor upregulated following ookinete invasion). These new data demonstrate that the number parasites (gametocytes, zygotes and ookinetes) in the midguts was similar across all conditions, however, whilst NF54-infected midguts showed significant upregulation of *LL3* and the new marker *SRPN6* due to ookinete invasion of the midgut epithelium as expected, the midguts containing DPY19-deficient parasites had much lower expression of both ookinete invasion markers. These data suggest that mutant ookinetes are not entering the epithelium as efficiently as NF54, in support of our hypothesis that CTRP is not secreted appropriately without C-mannosylation, resulting in ookinetes with reduced motility. Similar conclusions have been drawn from other *Plasmodium* ookinete studies using these markers (PMID 16260729, 18005239, 23093936, 23658788, 29873127). We have toned down our conclusion sentence as follows. “This suggests that DPY19-deficient ookinetes are not able to invade the midgut epithelium as efficiently as NF54 or that mosquito immune responses cannot recognize these mutant ookinetes effectively.”.

Fig. 5f: If normalized to aldolase as written on the y-axis, how come that the GFP expression is in both cases 100.0?

Response: In these experiments, we have used densitometry to measure the intensity of GFP-specific bands and standardised this to the loading control, Aldolase. First, we compared the ER-Signal cleaved GFP band between parasite lines relative to Aldolase. Separately, we compared the GFP core band between parasite lines relative to Aldolase. To allow visual comparisons between parasite lines, we set the control lane NF54 to 100% to reveal the degree of reduction in the Δ DPY19 lane. This is a standard methodology eg (PMID 24983235, 26832821).

The statement in line 320 is unclear and potentially wrong. TRAP mannosylation appears to depend on DPY19, but that does not necessary mean that there are no other C-mannosyltransferases across the life cycle. Please revise.

Response: The statement has been softened, as follows. “Collectively, these results establish DPY19 as a C-mannosyltransferase in *P. falciparum*.”

Line 392: For clarity, should it not be C-mannosylation?

Response: This has been corrected. “*P. falciparum* mutants lacking O-fucosylation³⁶ or C-mannosylation (this study) are defective for oocyst production in mosquitoes.”

Line 431: The data providing evidence that “DPY19 activity in the early secretory pathway is important” are very scarce and thus statement is too bold.

Response: We have removed ‘from the early secretory pathway’ from the sentence.

Reviewer #3 (Remarks to the Author):

The study by Lopaticki is a thorough and well-thought through exploration of C-mannosylation in the malaria parasite, which up until recently was not thought to possess glycosylation processes at all. Building on the work from the same group (having proven O-linked glycosylation) the team now demonstrate that C-mannosylation occurs and, perhaps more importantly, shows strong evidence that it is mediated by a homologue of the C-mannosyltransferase DPY19. The work uses a suite of technologies and appears (to this reviewer) as definitive in its proof that DPY19 in *P. falciparum* is responsible for C-mannosylation of key TSR domain containing proteins that are critical for transmission of the parasite through the mosquito.

The narrative of the discovery work I find clear and generally compelling. My only major issues are where there is a slight tendency to skip over experiments that, on their own, are only partially supportive of the conclusions being stated, but on the whole (together with everything else) do indeed still stand up. For example several key controls (negative controls) I would expect that are lacking. I would expect some of these should be corrected for completeness and to add weight to the overall arguments made.

Detailed comments:

1. Introduction L53: Have the authors looked at *Chromera velia* - a possible evolutionary ancestor to Apicomplexa?

Response: There probably is a DPY19 homolog and proteins with TSRs in members of the Chromerida phylum. For example, in *Vitrella brassicaformis* there is a putative DPY19 (<https://www.uniprot.org/uniprot/A0A0G4FVD8>) and proteins with TSR domains (<https://www.uniprot.org/uniprot/A0A0G4EVM5>). At the same time, the phylogeny of these organisms isn't clear cut, and the Chromerida are sometimes classified as a member of Apicomplexa (see <https://en.wikipedia.org/wiki/Alveolate>). Given this, and that the aforementioned *Vitrella* genes are yet to be characterised, we have amended the introduction to state: “Apicomplexan protists are the only single-celled organisms known to produce proteins with TSR domains”. This would appear to be factually correct at this time.

2. Results L97: I am skeptical whether alpha-fold 2 really adds anything here. My inclination would be to drop this as it really doesn't add anything to the paper i.e. speculative. A follow up perhaps but for now since none of this is explored I'd drop.

Response: Alphafold 2 (AF2) is a new tool to help predict protein domain boundaries. The reviewer is right that it is just a prediction and that it does not necessarily impact the findings of our manuscript, however we don't think that it is any more speculative than the use of MSA tools decades ago to make the original predictions of the protein's domain architecture. Indeed, the AF2 assignments provide a nice independent verification of these previous predictions, whilst also providing some new insights that may be of interest to those studying these important malaria proteins. Thus, we think it does have some value and is worth keeping. We have changed some of the wording in our results section to indicate that these are just predictions and indicate this more clearly in Fig 1a as well. We hope that this is an acceptable compromise for reviewer #3.

3. Results L116/F1f: I really feel that given the quality of the lightsheet microscopy this image is weak and doesn't support ER localisation. There is no negative control (to check this isn't a bad batch of anti-HA) and the distribution is not convincing. Is the ring even really a ring or a merozoite? The rest to me looks pretty much speckly i.e. uninformative. I'd suggest better images or if definitive proof of ER is required immuno-EM (though that is also quite hit and miss).

Response: We thank the reviewer for their comments. In metazoans, C-mannosylation is conferred to proteins in the ER (Hoffsteenge et al J Biol Chem 2001). To address this in *P. falciparum*, we have now prepared fresh DPY19-HA parasites and undertaken IFAs with the ER markers ER-calcium binding protein (ERC) and plasmepsin V (PMV). The ERC labelling was unfortunately problematic in these repeat experiments, perhaps due to degradation of the antibody stock, which we do not possess more of. However, IFA with the rabbit anti-PMV antibodies revealed the same perinuclear ER labelling for this protein that we and others have shown previously (Klemba et al Mol Biochem Parasitol 2005; Russo et al Nature 2010; Boddey et al Nature 2010; Sleebs et al PLoS Biology 2014) and the rat anti-HA signal in DPY19-HA was clearly shared with PMV in the cells we observed. To limit any potential issue with previous experiments or confusion about the subcellular localisation, which we feel is not central to the main conclusions of the paper, we have softened the conclusion to "most likely localized to the ER (see below)", removed the figure panels from the earlier version including other lifecycle stages and any reference to subdomains of the ER. We have also edited the results section as follows: "C-mannosylation of metazoan proteins is conferred in the ER⁴. To determine the subcellular localization of DPY19 in *P. falciparum*, transgenic NF54 parasites were produced, whereby the *DPY19* gene was tagged at the C-terminus with a triple hemagglutinin (HA) epitope. Integration of the HA epitope was demonstrated by Southern blot (Supplementary Fig. 2a) and successful tagging confirmed by immunoblot (Fig. 1e). Immunofluorescence microscopy revealed a perinuclear localization for DPY19-HA that co-localized with the ER-resident protease plasmepsin V³⁸ in ring-stage parasites (Fig. 1f). The distribution pattern for DPY19-HA indicates that this enzyme is most likely localized within the ER, similar to metazoans."

4. Results L137: A question more than a comment - why is DPY19 there if its is dispensable? Could other conditions be explored to show it does serve a blood stage function? Shaking cultures? Nutrient deprivation? This is a recurring problem in asexual KOs, absence of evidence isn't evidence of absence!

Response: We agree with the reviewer that is curious that asexual blood stages of *P. falciparum* and *P. berghei* actively express the O-fucosylation and C-mannosylation pathways

yet, under the experimental conditions used, no detectable phenotype has been identified in corresponding pathway knockouts by us or others (PMID 28916755, 31334132, 31559936, this study). It is possible that the growth conditions tested apply insufficient selective pressure as suggested by the reviewer, however, answering this question, whilst interesting, is beyond the scope of this study.

5. Results L161: I don't understand how to interpret Figure 3a. The reference to it says rSPATR is discussed in this sentence but the blot ONLY shows rMTRAP as showing up? Is this mis-labeled or am I mis-interpreting what is being shown/described.

Response: We have repeated these experiments a little differently, and hopefully it is now easier to interpret. Please see comments to reviewer 2 for more detail.

For example, we have repeated the experiment in Fig 3a in a slightly different way to address the reviewers' concerns and updated the figure, caption and text accordingly. We induced *P. falciparum* TSR protein secretion in six of our yeast lines (no vector, SUMO-MTRAP, SUMO-SPATR: all +/- constitutive co-expression of CeDPY19) at equivalent cell densities. An equal volume of each culture supernatant was used to perform new Western blots (see revised Fig 3a). We first probed the membrane with anti-FLAG to measure expression of each TSR protein: this indicated robust expression of the FLAG-SUMO-MTRAP fusion protein in the presence or absence of CeDPY19. By contrast, very little FLAG-SUMO-SPATR was obtained without co-expression of CeDPY19, likely because the protein requires this modification for stability and/or secretion. The same membrane was then probed with 5G12 (anti-EWMan): this confirmed C-mannosylation of MTRAP when CeDPY19 was co-expressed. Since SPATR possesses a DW(Man) in place of EW(Man) in MTRAP that is recognized by 5G12, rSPATR was not recognized by the antibody as expected, demonstrating specificity of the antibody. Nonetheless, mass spectrometry confirmed that rSPATR was indeed C-mannosylated at the DW(Man) epitope but only when co-expressed with CeDPY19 (Supp Fig 4).

6. Results L180: I would avoid terminology like "remarkable". Perhaps qualify expected and unexpected.

Response: This section has been revisited, and now reads "*Differential scanning fluorimetry (nanoDSF) was used to assess the stability of rMTRAP with and without C-glycosylation. The C-glycosylated sample exhibited a bulk $T_m \approx 6$ °C higher than the unglycosylated sample (Fig. 3d), even though glycosylation occupancy in this sample was modest ($\approx 20\%$). This indicates that C-mannosylation confers considerable additional stability to the MTRAP protein fold.*"

7. Results L189/Figure 3e: I'd really like a negative control here, immuno IPs are tricky and showing the banding pattern looks ok but this can be just heavy and light chains. I'd like to see the same blot without an IP ab (show that there's nothing x-reactive) and also a pre immune serum (or unrelated ab) that shows no x-reactivity. This would make it more convincing and less prone to being a mis-interpreted result.

Response: We understand and share the caution urged by the reviewer. Unfortunately, the blots and the MTRAP-HA parasite lysates (from 30x 30 ml dishes) were discarded and, owing to a competing study on C-mannosylation from a competing laboratory currently in the process of publication, we have not been able to repeat this. However, we provide further explanation of the results for the reviewer. First, the MTRAP-HA bands detected (p58 and p35) are the same as detected previously (see PMID 32109369, Figure 6F). We have

included a statement to this effect, as follows. “HA antibodies specifically identified MTRAP-HA as two protein species, p58 and p35, as previously described⁴².” Second, probing the same membrane with 5G12 antibodies identified that the same bands contained the modified EW(Man) epitope(s) (Fig. 3e). Third, the sizes of heavy (~55 kDa) and light (~25 kDa) chains are significantly different to those observed in our immunoblots of ~70 kDa and ~40 kDa (called p58 and p35, respectively, in PMID 32109369). We have compromised by softening the conclusion by changing the wording from ‘unambiguously demonstrating’ to “strongly suggesting”.

8. Results L225: The movies are amazing, but there is SO much heterogeneity in exflagellation events to my (admittedly untrained eye) I find Suppl. Movie 4 not entirely dissimilar from WT that I have seen.... i.e. Is this really a mutant phenotype?

Response: We appreciate the candour and question from the reviewer. The exflagellation occurring in Supp Movie 4 by Δ DPY19 is occurring from completely within the erythrocyte ie., the residual body remains inside the host cell and no erythrocyte unwrapping was evident. In contrast, this type of exflagellation was not observed in any of the movies of NF54 controls, where the gametocytes either i) completely egressed the host cell soon after activation, or ii) unwrapped the erythrocyte membrane and partly egressed before exflagellation occurred. This is probably linked to the partial MTRAP phenotype in Δ DPY19.

9. Results L244: Are the authors not surprised they didn't see the same phenotype as the MTRAP KO - i.e. a super tadpole like bundle of flagellum in unruptured RBC/PVM? I was expecting something similar to this but it seems this is not what was seen? I'd be keen to see this commented on.

Response: We thank the reviewer for this question. We think that the different phenotype is due to only partial knockdown of MTRAP in Δ DPY19 ie. some of this protein remains. Thus, residual MTRAP is providing a different phenotype compared to the study on Δ MTRAP (PMID 27832590) where the protein pool is completely abolished. A comment has now been added to the Discussion as follows. “Interestingly, gamete egress by Δ DPY19 did not completely phenocopy that reported for Δ MTRAP¹⁸. The different phenotypes are probably due to only partial knockdown of MTRAP in Δ DPY19, whereas in the study with Δ MTRAP, the protein was completely abolished¹⁸.”

10. L307/Figure 6b: There's a danger here that we are looking at N=1 sporozoites for the KO. Is there any way of increasing the numbers shown to really demonstrate the 5G12 staining of KO parasites is actually really (consistently) different?

Response: We thank the reviewer for their concern and also share this strongly when reading the literature as images can be cherry picked. We have shown images that represent multiple sporozoites, however we now provide additional images of sporozoites (NF54, Δ DPY19 and Δ DPY19 comp) which are admittedly very hard to produce because of the magnitude of the transmission defect, to demonstrate the results. They all showed the same phenotype. The additional sporozoite images confirming similar observations that Δ DPY19 sporozoites do not label like the NF54 and the DPY19 complemented line are now presented in Supp Fig. 11.

Reviewer comments, second round review: –

Reviewer #2 (Remarks to the Author):

In the revised version of the manuscript, the authors corrected the mistakes from the initial submission and added some new data, e.g., Fig. 3a and Fig. 4i. The revision appears sufficient and the manuscript clearly benefitted from the removal/toning down of some statements, which were based on relatively weak data.

At times, some of the additional information, which was provided in the response to the reviewer, would have strengthen the manuscript (e.g., that Fig. 3d shows duplicate melt curves of recombinant MTRAP samples grown in presence or absence of CeDPY19).

Fig. 4 and corresponding text. Although improved in the revised version, the description of the various egress and exflagellation phenotypes in wild type and mutant is still difficult to approach. I still find it artificial to talk about "macrogamete egress", despite what is shown in PMID 27111866. Why not referring to a female parasite as macrogametocyte as long as it is inside the host cell and a macrogamete once it is out?

Similarly, using the term "microgamete egress" in Fig. 4j instead of the established term "exflagellation" is confusing. The term "axoneme egress" (line 221) is wrong as the axoneme normally stays within the parasite. The term "egress" should be limited to the event of a parasite leaving the host cell and "exflagellation" should describe the emergence/budding of male gametes from the male gametocyte.

Reviewer #3 (Remarks to the Author):

The authors have made a very fair attempt to address all of my concerns, with inclusion of new data or clarification. No study is definitive, but the authors should be commended for giving it a fair go! I have no further comments/concerns.

Reviewer #2 (Remarks to the Author):

In the revised version of the manuscript, the authors corrected the mistakes from the initial submission and added some new data, e.g., Fig. 3a and Fig. 4i. The revision appears sufficient and the manuscript clearly benefitted from the removal/toning down of some statements, which were based on relatively weak data.

At times, some of the additional information, which was provided in the response to the reviewer, would have strengthened the manuscript (e.g., that Fig. 3d shows duplicate melt curves of recombinant MTRAP samples grown in presence or absence of CeDPY19).

Response: We have now added further information for clarification. First, that duplicate samples were analysed is now reflected in the main text using pluralization: “The C-glycosylated samples exhibited a bulk $T_m \approx 6$ °C higher than the unglycosylated samples (Fig. 3d)”. Second, the Figure 3d legend has been updated as follows: “Thermal denaturation profiles and T_m values for rMTRAP with (dashed lines) and without (hard lines) C-mannosylation, as determined by nanoDSF. Duplicate curves for each condition are shown.”.

Fig. 4 and corresponding text. Although improved in the revised version, the description of the various egress and exflagellation phenotypes in wild type and mutant is still difficult to approach. I still find it artificial to talk about “macrogamete egress”, despite what is shown in PMID 27111866. Why not referring to a female parasite as macrogametocyte as long as it is inside the host cell and a macrogamete once it is out?

Response: We have changed “macrogamete egress” to “macrogametocyte egress”. We have also edited the text as follows: “Analysis of successful gametogenesis events by NF54 and $\Delta DPY19$ revealed no significant differences in timing of key developmental events between parasite lines (Fig. 4j). Collectively, lattice light-sheet imaging revealed that both males and females use DPY19 to egress the erythrocyte during gametogenesis and that exflagellation can originate from a residual body located outside or sometimes inside the host erythrocyte.

Similarly, using the term “microgamete egress” in Fig. 4j instead of the established term “exflagellation” is confusing. The term “axoneme egress” (line 221) is wrong as the axoneme normally stays within the parasite. The term “egress” should be limited to the event of a parasite leaving the host cell and “exflagellation” should describe the emergence/budding of male gametes from the male gametocyte.

Response: We have removed the terms “microgamete egress” and “axoneme egress”.

Reviewer #3 (Remarks to the Author):

The authors have made a very fair attempt to address all of my concerns, with inclusion of new data or clarification. No study is definitive, but the authors should be commended for giving it a fair go! I have no further comments/concerns.

Response: We thank the reviewer for their supportive feedback.